

# SAFE-CAST: secure AI-federated enumeration for clustering-based automated surveillance and trust in machine-to-machine communication

Yusuf Kursat Tuncel and Kasım Öztoprak

Department of Computer Engineering, Konya Food And Agriculture University, Konya, Turkey

## ABSTRACT

Machine-to-machine (M2M) communication within the Internet of Things (IoT) faces increasing security and efficiency challenges as networks proliferate. Existing approaches often struggle with balancing robust security measures and energy efficiency, leading to vulnerabilities and reduced performance in resource-constrained environments. To address these limitations, we propose SAFE-CAST, a novel secure AI-federated enumeration for clustering-based automated surveillance and trust framework. This study addresses critical security and efficiency challenges in M2M communication within the context of IoT. SAFE-CAST integrates several innovative components: (1) a federated learning approach using Lloyd's K-means algorithm for secure clustering, (2) a quality diversity optimization algorithm (QDOA) for secure channel selection, (3) a dynamic trust management system utilizing blockchain technology, and (4) an adaptive multi-agent reinforcement learning for context-aware transmission scheme (AMARLCAT) to minimize latency and improve scalability. Theoretical analysis and extensive simulations using network simulator (NS)-3.26 demonstrate the superiority of SAFE-CAST over existing methods. The results show significant improvements in energy efficiency (21.6% reduction), throughput (14.5% increase), security strength (15.3% enhancement), latency (33.9% decrease), and packet loss rate (12.9% reduction) compared to state-of-the-art approaches. This comprehensive solution addresses the pressing need for robust, efficient, and secure M2M communication in the evolving landscape of IoT and edge computing.

## INTRODUCTION

The contemporary technological infrastructure largely relies on machine-to-machine (M2M) communication, which allows devices to interact directly with one another without the need for human supervision. This model of operation is finding its way into many new applications, from active control and automation to the not-so-novel field

Corresponding author
Yusuf Kursat Tuncel,
kursat.tuncel@gmail.com

of passive monitoring systems (*Prabhakara Rao & Satyanarayana Murthy, 2023*). Yet, with the increasing ubiquity of M2M, a growing number of security problems has come to light. The primary security risks inherent in M2M communication include:

- **Physical tampering:** Unauthorized physical access to M2M devices can lead to data breaches or device manipulation.
- **Unauthorized monitoring:** Interception of M2M communications can compromise sensitive data and privacy.
- **Hacking:** Malicious actors may exploit vulnerabilities in M2M systems to gain unauthorized control or access to data.
- **Insecure M2M connections:** The heterogeneity of systems and technologies deployed in M2M devices contributes to substantial vulnerabilities, complicating the implementation of effective defensive measures (*Bilami & Lorenz, 2022*).
- **Large-scale sensitive data leakage:** The vast amount of data transmitted in M2M networks increases the risk of large-scale data breaches (*Moussa et al., 2022*; *Luo et al., 2023*).
- **Inadequate surveillance:** Lack of effective monitoring systems makes it challenging to detect and respond to security threats in real-time.
- **Compromised Quality of Service (QoS):** Security vulnerabilities can lead to disruptions in service, affecting the reliability and performance of M2M systems.

The security landscape is further complicated by the absence of unified communication standards and the proliferation of vulnerable devices across the expanding Internet ecosystem. This fragmentation of protocols and the surge of poorly protected endpoints amplify existing security risks, creating a more complex and challenging environment to safeguard M2M communications. As a result, the proliferation of M2M systems could potentially introduce a significant number of vulnerable points in the wider network ecosystem (*Dehalwar et al., 2022*; *Zukarnain, Muneer & Ab Aziz, 2022*; *Santhanakrishnan et al., 2022*).

To address these challenges, researchers have introduced various strategies aimed at improving the security of M2M communication. For instance, *Panda, Mondal & Kumar (2022)* proposed a secure and lightweight authentication protocol (SLAP) for M2M communication in Industry 4.0, focusing on ensuring robust security measures with minimal computational overhead. *Djehaiche et al. (2023)* discussed adaptive control of Internet of Things (IoT)/M2M devices in smart buildings using heterogeneous wireless networks, highlighting the need for secure and efficient communication protocols. Similarly, *Ghasri & Hemmatyar (2022)* introduced a dynamic optimal M2M radio frequency (RF) interface setting for IoT applications with the aim of improving the security and reliability of M2M communications.

Other notable contributions include the work of *Shahzad et al. (2022)*, who developed a single-factor lightweight authentication protocol (SF-LAP) for secure M2M communication in industrial IoT (IIoT), highlighting the importance of low power and efficient security solutions. *Kaushal et al. (2022)* proposed a secure IoT framework for medical applications, addressing the critical need for secure data transmission in healthcare

settings. *Bilami & Lorenz (2022)* presented a lightweight blockchain-based scheme to secure wireless M2M area networks, focusing on data integrity and secure storage (*Choudhary & Pahuja, 2023*; *Nyangaresi, Rodrigues & Abeka, 2023*).

Despite these advancements, existing solutions remain susceptible to various issues. The heterogeneity in M2M communication, coupled with the lack of effective machine clustering and the selection of appropriate cluster heads (CHs), leads to increased power consumption. Furthermore, only a few existing approaches incorporate security measures during data transmission, resulting in increased risks of data privacy breaches. The deficiency in effective surveillance and maintenance further compromises the security of M2M communications.

The overall goal of this research is to significantly reduce power consumption, minimize data leaks in edge-assisted M2M communication, and enhance overall system security. To achieve this, we propose a novel approach: federated secure clustering-based communication enhanced with trust enumeration, surveillance, and maintenance for enhanced security analysis. This method is designed to boost security, thereby minimizing packet loss and power consumption.

Our approach addresses the following key challenges:

- **Amplification of data privacy:** Through secure clustering and dual cluster head selection.
- **Minimization of sensitive data leakage:** By implementing secure channel selection and strategic channel divisions.
- **Securing M2M communication:** Using enhanced blockchain technology for improved scalability and privacy in data transmission.
- **Enumeration of machine trust:** Estimating trust based on significant parameters to establish a secure communication environment.
- **Enhancement of surveillance and maintenance:** Efficiently detecting communication threats and anomalies and accurately predicting link states to reduce packet loss.

This research includes both theoretical analysis and experimental demonstrations to evaluate efficiency and security. The theoretical analysis provides insight into the expected performance and security enhancements of the proposed method, while experimental demonstrations validate these theoretical predictions under practical conditions, ensuring a comprehensive solution to the inherent vulnerabilities of M2M communication systems.

The primary reason for the development of the secure AI-federated enumeration for clustering-based automated surveillance and trust (SAFE-CAST) framework was the need to provide more secure and efficient M2M communication systems. Without these two requirements being met concurrently, one can have an insecure system or an inefficient system. Existing solutions often prioritize one aspect at the expense of the other, leaving systems either vulnerable to security breaches or suffering from under-valued performance. SAFE-CAST aims to close this gap by providing a comprehensive framework that ensures hardened security measures without compromising on system efficiency, thereby addressing the fundamental challenge of balancing protection and performance in M2M networks. Many current methods concentrate on single security or effectiveness matters while

ignoring holistic threats or only working intermittently as needed by an organization. A fully integrated framework of SAFE-CAST, coherent to channel selection, among others, is made up of federated learning, quality-diversified optimization, blockchain-based trust management, and adaptive reinforcement learning, among others. Efficient energy usage and improved network performance will be achieved, which will also ensure that all levels of M2M communication are secure in this context.

The rest of the paper is structured as follows: 'Literature Survey' presents a literature survey on existing M2M security solutions, highlighting current approaches and their limitations. 'Problem Statement And Existing Challenges' outlines the problem statement and existing challenges in M2M communication security. 'Proposed Method' details our proposed methodology, SAFE-CAST, including its key components: federated clustering, secure channel selection, trust evaluation, and surveillance mechanisms. 'Analysis of Key Algorithms in SAFE-CAST' provides a theoretical analysis of the SAFE-CAST efficiency and security aspects. 'Experimental Results' presents our experimental results, including simulation setup, comparative analysis, and research findings. Finally, 'Conclusion and Future Work' concludes the paper, summarizing key contributions and suggesting future research directions.

## LITERATURE SURVEY

This section critically reviews the shortcomings of previous research. One study (*Railkar, Mahalle & Shinde, 2021*) proposed a fuzzy-based trust score estimation method for M2M communication with the aim of creating a scalable trust management (STM) paradigm. Although STM offers a structured approach, its effectiveness is limited when trust assessment is based on a narrow set of parameters, potentially rendering the estimates less reliable. Another research (*Zhang et al., 2021*) introduced an efficient and privacy-preserving blockchain security solution for online social networks, integrating keyword search strategies with public-key encryption and blockchain technology. This method offers an effective keyword search mechanism for data queries, but the lack of the necessary verification processes can be a drawback in certain scenarios.

In M2M communication, effective key-selection techniques often require extensive computational resources, leading to increased processing overhead. A different study (*Weng et al., 2019*) presented the DeepChain incentive mechanism along with a collaborative training paradigm, where parties share local gradients for collective deep learning (DL) training. However, this approach used a standard blockchain, which is known to have scalability issues in data storage and transactions.

In another paper, researchers (*Sanober et al., 2021*) proposed an enhanced DL system to detect fraud in wireless communications. It modified principal component analysis (PCA) for relevant feature selection but faced challenges with interpretability and increased complexity. Another research (*Fatani et al., 2021*) developed a DL-based efficient intrusion detection system using the Transient Search Optimization (TSO) technique. The approach involved feature extraction using convolutional neural networks (CNN) and feature selection modified by TSO, known as TSODE, employing the differential evolution (DE)

algorithm's operators. Although effective, the CNN algorithm can generate numerous extraneous layers during processing, which might result in increased latency. Further investigations in the literature reveal a diverse array of approaches and challenges in M2M communication and IoT security.

Another study (*Mazhar et al., 2022*) suggested an intelligent forensic analysis strategy for the detection of M2M-based automated attacks on IoT devices. Here, many machine learning (ML) algorithms and forensic analysis tools were used to construct the M2M framework to identify the kind of assault. The method makes use of many ML techniques, including the decision tree (DT) algorithm, which works with a high degree of accuracy to identify attacks automatically. The DT technique was used in this scenario to identify attacks. However, overfitting occurs because of the algorithms' lack of sample dependency. In a recent research (*Xu et al., 2022*), a unique bidirectional linked blockchain (BLB) was enabled for assault defense. Here, the bidirectional references between blocks were created using the suggested chameleon hash function. At the same time, a novel consensus process known as the committee member auction (CMA) was created to improve security and resilience to BLB attacks, thereby achieving great scalability. Bidirectional linked blockchains have scalability issues, particularly in situations where a lot of devices are connected. The research carried out in *Ejigu & Santhosh (2020)* presented an M2M communication-based autonomous home automation and security system. To operate, monitor, and manage appliances remotely, the suggested work adapts the merging of wireless communication, the cloud, and services. All of the users would converse with each other in this situation. M2M communication is facilitated by the message queueing telemetry transport (MQTT) messaging protocol. For M2M communication, the MQTT protocol was used. However, the absence of data encryption raises the risk of hostile activity. *Samy et al. (2021)* suggested improving IoT security by using an efficient protocol for M2M authentication. In the beginning, devices were registered by giving their details. After that, secret key generation in the key agreement phase was done using the elliptic curve cryptography (ECC) technique. In addition, the public key cryptography (PKC) process was used to enhance security. The secret key was produced using the ECC technique, its encryption size was noticeably larger, and secure establishment was challenging. *Al-Shareeda et al. (2022)* proposes to analyze replay attacks that may occur in the SECS/GEM system. The primary goal of the proposed work is to defend SECS/GEM communications from replay assaults. Here, the threat that threatens the existing state of the operation-based control system was revealed. Due to binary encoding, SECS/GEM does not provide any security features.

*Mahdavisharif, Jamali & Fotohi (2021)* focused on using long-short-term memory (LSTM) for attack detection, capitalizing on the algorithm's ability to maintain both long-term and short-term temporal information dependencies, but faced challenges due to the extensive training time required by LSTM, leading to increased latency. The study by *Wazid, Das & Shetty (2022)* proposed a trust aggregation certificate-based authentication method (TACAS-IoT) for secure communication in an edge-enabled IoT context, with the aim to safeguard the environment and detect various assaults, yet the approach did not sufficiently address device vulnerabilities, increasing the risk of malicious activities. In *Ahmed et al.*

*(2023)*, a lightweight authentication protocol-based authentication-chains protocol was suggested within a distributed decentralized blockchain ledger in an authorized IoT ecosystem, establishing cluster nodes and generating an authentication blockchain for each cluster, although its lightweight nature made it more susceptible to cryptographic attacks. The purpose of *Modiri, Mohajeri & Salmasizadeh (2022)* was to enhance M2M communication security through a group-based secure, lightweight authentication and key agreement (GSL-AKA) protocol, using automated validation of applications and protocols (AVISPA) to identify different types of attacks, but it could suffer from long delays and substantial communication and computational overhead. Lastly, *Jin et al. (2023)* proposed developing a historical data-entrenched multi-factor authenticated and confidential channel (HMACCE), validated using the secret key managed by IIoT and storing historical data and tags in server relationships, but despite improved security, the approach still faced significant breaches due to a lack of thorough vulnerability analysis.

In a following study (*Umran et al., 2023*), an architecture-based private blockchain network/smart contract and interplanetary system were presented, focusing on security, scalability, speed, decentralization, privacy preservation, and reliability. The approach enhanced blockchain functionality using the Merkle tree's incremental aggregator subsector commitment and consensus algorithms as multi-chain evidence for rapid authentication. However, the complexity of maintaining and implementing these consensus techniques poses significant challenges for organizations when integrating them into their existing authentication systems. *Li et al. (2023)* proposed an editable blockchain-based IIoT device authentication method suitable for large-scale scenarios, addressing the issue of low energy consumption in devices. A lightweight identity authentication protocol, BLMA, was developed to tackle authentication challenges between industrial devices. The system relies on validate-practical Byzantine fault tolerance (vPBFT), which depends heavily on message forwarding between nodes for consensus. As the number of nodes increases, so does the communication cost, affecting the system's speed and scalability. The goal of the study by *Sasikumar et al. (2023)* was to enhance the security and efficiency of IoT systems by introducing a novel approach to decentralized resource allocation in edge computing environments, with the caveat that private information stored on IoT devices is vulnerable to hacking. In *Gupta et al. (2023)*, a lightweight authentication mechanism was implemented for multiple enhanced machine-type communication (eMTC) devices in the 5G ecosystem using a group-leader approach. Although offering increased security, stronger authentication techniques surpass the security level of this lightweight approach. Table 1 outlines the research gaps identified in the literature survey.

The following paper, *Alsultan, Oztoprak & Hassanpour (2016)* discusses advancements in wireless sensor networks (WSNs), focusing on energy-efficient and scalable routing protocols. It introduces the multi-hop, far-zone, and load-balancing hierarchical-based routing algorithm (MFLHA) to address the limitations of traditional routing algorithms, such as inefficient energy consumption and reduced network lifetimes. MFLHA improves network performance by prioritizing higher-energy nodes as CHs, creating a far-zone for energy-efficient sensor communication, and implementing a multi-hop inter-cluster routing algorithm to reduce energy consumption by CHs, ultimately enhancing the

**Table 1  Research gap in literature survey.**

| Reference | Objectives | Methods or algorithms used | Limitations |
|---|---|---|---|
| *Railkar, Mahalle & Shinde (2021)* | To develop the M2M communication paradigm of scalable trust management (STM) | Fuzzy approach | A limited set of security parameters. |
| *Zhang et al. (2021)* | To provide a blockchain security solution for online social networks that is both effective and private-preserving | Efficient keyword search algorithm | Increases processing overhead. |
| *Weng et al. (2019)* | To build a DeepChain prototype and test it in various scenarios on an actual dataset | Deep chain mechanism | Scalability issues. |
| *Sanober et al. (2021)* | To propose a DL system with enhanced security for detecting wireless communication fraud | Random Forest, Support Vector Machine (SVM), K-Nearest Neighbor, Logistic regression (LR), DT | Increase in complexity. |
| *Fatani et al. (2021)* | To introduce an effective AI-based method for IoT systems' intrusion detection systems (IDS) | Convolutional neural networks (CNNs), Transient Search Optimization (TSO) algorithm | Increased latency. |
| *Mazhar et al. (2022)* | To put forth a method for M2M-based automatic threat detection in IoT devices using intelligent forensic analysis | DT, RF, Naïve Bayes (NB) | Overfitting problems. |
| *Xu et al. (2022)* | To implement a special bi-directional linked blockchain (BLB) for attack defense | Committee members auction (CMA) consensus algorithm | Scalability issues. |
| *Ejigu & Santhosh (2020)* | To provide voice-assisted, IoT-based, cross-platform, simple, easy, versatile, multi-way, comprehensive, user-friendly, and self-governing methods for monitoring and managing household appliances and security systems | Message Queuing Telemetry Transport (MQTT) | Increases the possibility of hostile activity. |
| *Samy et al. (2021)* | To enhance IoT security by using a successful M2M authentication mechanism | Optimized protocol | The encryption size was larger, and secure setup was difficult. |
| *Al-Shareeda et al. (2022)* | To prevent replay attacks on SEC-S/GEM communications | Binary-encoded communications | Does not provide any security measures. |
| *Mahdavisharif, Jamali & Fotohi (2021)* | To identify attacks using long short-term memory (LSTM), where the algorithm can maintain both short- and long-term temporal relationships of data | Big Data-Deep Learning IDS (BDL-IDS) | Increased latency. |
| *Wazid, Das & Shetty (2022)* | To enable secure communication in an IoT setting by using a trust aggregation-certificate-based authentication method | TACAS-IoT | Raised the risk of malicious behavior. |
| *Ahmed et al. (2023)* | To reduce resource usage and maintain a decentralized authentication procedure in the IoT | Consensus algorithm | More vulnerable to cryptographic assaults. |

**Table 1** (*continued*)

| Reference | Objectives | Methods or algorithms used | Limitations |
|---|---|---|---|
| *Modiri, Mohajeri & Salmasizadeh (2022)* | To enhance the security of communication between M2M devices | GSL-AKA | Delay and computation overhead. |
| *Jin et al. (2023)* | To construct multi-factor ACES protocols using the random oracle model's historical data | HMACCE | One major cause of security breaches is inadequate vulnerability assessments. |
| *Umran et al. (2023)* | To provide an interplanetary system of architecture and a private blockchain network that is decentralized, quick, scalable, safe, private, trustworthy, and uses little power | Consensus algorithm | Consensus approaches are hard to maintain and use. |
| *Li et al. (2023)* | To deliver an editable blockchain-based IIoT device authentication system that can address the issue of low energy consumption in devices while meeting the requirements of large-scale scenarios | vPBFT algorithm, online and offline signature algorithm | An increase in communication costs affects the system's overall speed and scalability. |
| *Sasikumar et al. (2023)* | To present an innovative method for edge computing scenarios' decentralized resource allocation, therefore enhancing the security and effectiveness of IoT systems | The Secure Hash Algorithm (SHA)-256 | IoT devices that store personal data are vulnerable to hacking. |
| *Gupta et al. (2023)* | To implement a low-power authentication solution for several enhanced Machine Type Communication (eMTC) devices into the 5G network | Group-leader technique | Limited security. |

network's lifetime and efficiency. Similarly, there are studies trying to minimize energy usage while maximizing the capacity usage in wireless networks (*Kihtir et al., 2022*; *Oztoprak, 2018*).

One study by researchers investigates the protocol independent switch architecture (PISA), focusing on its application in enhancing distributed denial-of-service (DDOS) attack detection and empowering operators to develop network code (*Butun, Tuncel & Oztoprak, 2021*) independently. In another study (*Oztoprak, Tuncel & Butun, 2023*), researchers discuss the shift towards the edge-cloud continuum, particularly emphasizing the role of DevOps in network evolution and the integration of advanced technologies like AI/ML and edge computing. In addition, researchers also explore PISA's application in security traffic inspection, highlighting its role in efficient network management and the development of comprehensive security systems (*Oztoprak & Tuncel, 2023*). Together, these studies underscore the ongoing evolution and challenges in telecommunications, offering insights into future network technology and security directions.

# PROBLEM STATEMENT AND EXISTING CHALLENGES

## Problem statement

An ongoing key problem is to improve maintenance and surveillance-based secure online M2M communication. A few of the specific problems highlighted in the latest research are as follows:

*Lokhande & Patil (2021)* developed secure and energy-efficient M2M communication to enhance the energy efficiency of medical sensor nodes in telerobotic systems. An energy-efficient routing technique called event-driven duty cycling (EDDC) was suggested. To determine whether a node was an attacker or a regular node, an estimation of the periodic trust score for each node was made using three metrics. To achieve this goal, the degree of node (DN), energy available (EA), and successful packet delivery ratio (SPD) are assessed. In an other original study (*Shahzad et al., 2022*), communication security was guaranteed by establishing a secure authentication system. To facilitate secure M2M communication, the SF-LAP was enabled. A suggested protocol for analyzing the transmission between the sensor and controller modified the exclusive-OR operation and hashing function. Some of the problems detected in these papers are:

- The trust score was assessed for each computer inside the network to improve security. If the machines are not clustered, maintenance will be inefficient, and power consumption will be excessive.
- Although considering trust scores in terms of SPD, EA, and DN allowed for the successful identification of the attacker in the study, the accuracy of anomaly detection was harmed by depending only on trust score consideration.
- In addition, the study used trust estimation-based attack detection for energy-efficient communication. However, sensitive data leakage and data privacy are impacted by data transfer without any privacy concerns.
- The single-factor lightweight authentication protocol was used in this study to protect M2M connection, yet the attacker was able to tamper with data since there was no secure route for data transmission.
- This case included protecting the connection to avoid eavesdropping and desynchronization assaults, but the security of M2M communication is still impacted by the lack of thought given to potential attack scenarios.
- Moreover, AVISPA was implemented for informal and broad verification to guarantee security. The absence of monitoring and link state prediction leads to increased security failures and packet loss.

## Existing challenges in M2M communication security

The ES-SECS/GEM technique was introduced (*Laghari et al., 2023*) to bolster security in M2M communication within industrial networks. This model encompasses three critical stages: attack prevention, message integrity, and authentication. The hash-based message authentication code (HMAC) was employed to safeguard data transfers between machines. Another study (*Bilami & Lorenz, 2022*) presented a private blockchain-based lightweight scheme for authenticating M2M interactions, utilizing blockchain technology to enhance data security and ensure availability and traceability. The identification process was structured into three modules to solidify communication security: pre-registration, registration, and authentication. However, researchers encountered several challenges:

- To ensure security, the study used authentication code blocks for machine authentication. High-power consumption is an issue when clustering mechanisms are not employed.
- To safeguard data transit between computers and enhance privacy, a hash-based message authentication code was utilized. Nevertheless, data privacy and quality of service are compromised when data is not encrypted prior to transmission.
- The study carried out attack detection and prevention by validating the message's authenticity. However, in scenarios where intelligence is lacking, this approach adversely affects the quality of service and leads to ineffective detection of attacks.
- Despite the implementation of attack detection and prevention techniques to improve M2M communication security, there was a lack of ongoing maintenance and monitoring to support acceptable QoS.

These issues underscore the necessity for more robust and intelligent systems in M2M communication to enhance security, efficiency, and reliability.

- To improve security, the pre-shared key was exchanged over a secure channel, although the difficulty of making the whole conversation *via* a random channel increased significantly.
- Because there is a shortage of sensitive data, the standard blockchain has non-scalability problems. In this case, the lightweight blockchain was modified for safe data handling and transmission.
- The SHA-3 algorithm, which uses 256 bits for secret key creation and transmission in the study, is vulnerable to collision attacks and requires a long time for key generation, increasing latency and reducing security.

*Gong, Feng & Albettar (2022)* presented the PUF-based efficient authentication and session establishment (PEASE) protocol, an effective authentication and session establishment method based on physical unclonable functions (PUFs). Two message interaction mechanisms are included in the proposed protocol to provide strong availability and security. In this case, the primary goal of the suggested work was to improve security and availability concerns about edge machine connectivity in M2M communication. Some of the major problems in the study are:

- Whereas the fuzzy extractor finds it tough to handle feature space and entropy estimates, leading to complexity, it was used by the supervisor to ensure the machine's legitimacy.
- Although a secret key and master key were supplied here for safe communication, a deficiency in data security measures caused a significant amount of sensitive data to leak.
- Although a secure communication protocol was developed in the aforementioned study, the trustworthiness of each machine was not assessed, and the lack of communication monitoring restricts the quality of service.

## Existing solutions *vs.* SAFE-CAST

The primary security risks inherent in M2M communication include physical tampering, unauthorized monitoring, and hacking. These vulnerabilities are exacerbated by the lack

of communication trends, leading to a scenario where the proliferation of M2M systems could result in a significant number of unsecured devices within the Internet framework (*Prabhakara Rao & Satyanarayana Murthy, 2023*). Furthermore, the heterogeneity of the systems and technologies deployed in M2M devices contributes to substantial vulnerabilities, complicating the implementation of effective defensive measures against various attacks. Consequently, surveillance in M2M communication has emerged as a critical component in bolstering security. To address these challenges, previous researchers have introduced a range of strategies aimed at improving the security of M2M communication. A notable concern is the security of M2M devices, which are typically designed to operate autonomously, thus lagging behind human oversight and being inherently more vulnerable (*Bilami & Lorenz, 2022*). Various methodologies have been explored, including efficient authentication protocols, message integrity validation, attack detection, and prevention mechanisms. These approaches focus on the establishment of robust authentication protocols to verify the legitimacy of devices, often involving the transmission of secret keys through secure channels. The primary objective of these protocols is to ensure confidentiality and safety in communication (*Moussa et al., 2022*; *Luo et al., 2023*).

Moreover, the integrity of messages is scrutinized to assess security, employing techniques like authentication codes and pre-shared key-based integrity checks. Rapid and efficient identity authentication is facilitated by aligning pre-shared keys with identity data during key management (*Dehalwar et al., 2022*; *Zukarnain, Muneer & Ab Aziz, 2022*; *Santhanakrishnan et al., 2022*). Further studies have focused on identifying attackers by evaluating the trustworthiness of individual machines or devices, with mechanisms implemented to identify and prevent authenticity-compromising attacks. Ensuring reliable and secure communication, especially in dynamic and decentralized environments, is crucial to prevent data breaches and mitigate the risk posed by malicious threats that could disrupt standard operations and impact a wide range of technologies (*Choudhary & Pahuja, 2023*; *Nyangaresi, Rodrigues & Abeka, 2023*).

Despite these advancements, existing solutions remain susceptible to various issues. For example, heterogeneity in M2M communication, coupled with the lack of effective machine clustering and the selection of appropriate CHs, leads to increased power consumption. Furthermore, only a few existing approaches incorporate security measures during data transmission, resulting in increased risks of data privacy breaches. The deficiency in effective surveillance and maintenance further compromises the security of M2M communications.

To overcome these challenges, we propose a novel approach: secure federated clustering-based communication augmented with trust enumeration, surveillance, and maintenance for enhanced security analysis. This method is designed to boost security, thus minimizing packet loss and power consumption, presenting a comprehensive solution to the inherent vulnerabilities of M2M communication systems.

In Table 2, we summarized the differences between SAFE-CAST and other major frameworks discussed in the paper by *Kazmi et al. (2023)*.

**Table 2 Comparison of SAFE-CAST and emerging 6G security frameworks.**

| Aspect | SAFE-CAST | Quantum-safe cryptography | AI-driven security | Blockchain-based security |
|---|---|---|---|---|
| **Threat model** | Focuses on M2M communication in IoT, addressing unauthorized access, data breaches, and CIA (Confidentiality, Integrity, Availability). | Addresses potential future threats from quantum computing that could break current cryptographic systems, focusing on quantum-resistant algorithms. | Targets dynamic and adaptive threat detection, focusing on advanced persistent threats (APTs) and real-time attack mitigation in 6G networks. | Provides decentralized trust, focusing on eliminating single points of failure and securing data integrity and access control across distributed networks. |
| **Countermeasures** | Utilizes advanced encryption techniques, potentially blockchain for secure data transmission. Includes homomorphic encryption for privacy-preserving computation, enabling secure data processing without decryption. | Quantum-resistant algorithms, such as lattice-based, hash-based, and code-based cryptography, designed to withstand quantum attacks. | AI/ML algorithms for intrusion detection, anomaly detection, and automated response mechanisms. | Decentralized ledger systems, smart contracts, and secure data aggregation techniques to enhance trust and transparency. |
| **Authentication techniques** | Likely employs traditional mutual authentication, lightweight cryptographic methods suitable for IoT. | Focuses on secure key exchange and signature schemes that remain secure against quantum attacks. | AI-enhanced biometric authentication, behavior-based authentication, and continuous monitoring of authentication events. | Blockchain-based authentication using decentralized identity management and token-based systems for secure access control. |
| **Scalability** | Designed to be scalable for IoT networks, possibly leveraging edge computing or decentralized architectures. | Scalability is a challenge, with ongoing research into efficient implementation and integration with existing infrastructure. | Highly scalable, capable of handling large volumes of data and adaptive to real-time network changes. | Highly scalable due to decentralized nature; however, consensus mechanisms can impact performance in large-scale deployments. |
| **Innovation** | Focuses on secure communication protocols in current IoT networks, potentially incorporating blockchain and AI. Uses homomorphic encryption to ensure privacy while allowing computations on encrypted data, a cutting-edge approach in privacy-preserving technology. | Pioneering the development of cryptographic techniques that are resilient against future quantum threats. | Innovates through the integration of AI with security, enabling proactive threat detection and mitigation. | Innovates by decentralizing trust and enhancing transparency, with smart contracts automating security policies. |
| **Implementation** | Targeted for real-time, resource-constrained IoT environments, ensuring secure and efficient M2M communication. Privacy-preserving computations using homomorphic encryption are implemented to secure data even during processing. | Implementation is still in research phase, with gradual adoption expected as quantum computing becomes a tangible threat. | AI-driven security solutions are being implemented in pilot projects, with a focus on adaptive security in real-time. | Already seeing adoption in various sectors, particularly in supply chain management, finance, and identity verification. |

**Table 2** (*continued*)

| Aspect | SAFE-CAST | Quantum-safe cryptography | AI-driven security | Blockchain-based security |
|---|---|---|---|---|
| **Research directions** | Focuses on immediate application and enhancement of current security protocols in IoT, including the exploration of homomorphic encryption for broader use cases in secure, privacy-preserving computations. | Extensive research into quantum-resistant algorithms and integration with current systems without significant performance loss. | Research is focused on enhancing AI's accuracy in threat detection, reducing false positives, and integrating AI with other security mechanisms. | Research is aimed at improving scalability, reducing energy consumption, and integrating blockchain with AI and IoT for comprehensive security solutions. |
| **Flexibility** | Adaptable to current IoT network conditions, with potential edge computing integration. The use of homomorphic encryption adds flexibility in handling sensitive data securely. | Less flexible due to the computational complexity of quantum-safe algorithms, but essential for future-proof security. | Extremely flexible, with the ability to adapt to new threats and network conditions dynamically. | Offers flexibility in secure, decentralized data management, but may face challenges in integration with legacy systems. |

## Research contributions

This study makes several key contributions. First, it develops SAFE-CAST, a framework that integrates various components of AI to create a more secure and efficient M2M communication environment. The second contribution is an implementation of a federated learning approach within the SAFE-CAST framework. This allows us to harness the learning potential of many "agents" -in this case, the many devices that make up an artificial IOT (AIoT)—without compromising their individual data privacy and security.

The QDOA for selecting channels in a secure and efficient way in M2M networks is another original contribution of this study. The creation of a trust management system among M2M devices is critical to the security of the network. Trust management is a problem that has been solved in many ways; this solution leverages the power and transparency of blockchain technology.

The innovative scheme that we named AMARLCAT minimizes latency and improves scalability in M2M communication. It is a new way of working that optimizes the adaptive multi-agent reinforcement learning algorithm and allows us in this lab to harness the power of the many communication events that occur within our communication system. The comprehensive performance evaluation that has been done has shown us how well this scheme works compared to existing methods. It certainly appears to be a great improvement over those methods, as it has shown significant improvements over what was there before in several key areas.

Dealing with the all-important necessity for strong, effective, and safe M2M communication systems in the ever-changing world of the Internet of Things and edge computing is the real-world consequences resulting from the output of this study.

## PROPOSED METHOD

In this research, we have mainly focused on enhancing privacy and security in online communication through surveillance M2M communication. In addition to that, the

intention of the proposed work is to evade resource constraints, latency, insecure communication, and packets. Here, we have utilized 6G communication to improve the speed and reliability of data communication, and enhanced blockchain is employed for secure data transmission and storage. This work embraces several entities, such as machines, edge servers, cloud servers, and invigilator agents (IA). Figure 1 describes the proposed architecture. The proposed work consists of four sequential processes,

- Federated clustering
- Unassailable channel selection & channel division
- Trust evaluation & secure communication
- Surveillance & maintenance

## Federated clustering
### *Process of federated clustering*
Federated clustering involves the following key steps:

1. **Local clustering:** Each machine, or node, in the network performs local clustering based on its observed data and interactions. This step involves grouping neighboring nodes that frequently communicate and exhibit similar trust values and behaviors. Local clustering algorithms, such as K-means or DBSCAN, can be employed for this purpose.

2. **Cluster head selection:** Within each local cluster, a CH is selected. The CH is responsible for aggregating data from its cluster members and communicating with other CHs. The selection of CHs can be based on criteria such as trust values, computational capabilities, and energy levels to ensure that the most suitable nodes take on this role.

3. **Federated aggregation:** CHs participate in a federated learning process, where they aggregate the data and models from their respective clusters. This step involves sharing and updating a global model without transferring raw data, thereby preserving data privacy and reducing communication overhead.

4. **Global model update:** The aggregated models from CHs are used to update a global model, which is then distributed back to the CHs. This global model helps maintain consistency and improve the overall performance of the network.

5. **Iterative refinement:** The process is iterative, with multiple rounds of local clustering, aggregation, and global model updates. Each iteration refines the clusters and improves the accuracy and efficiency of the overall system.

### *Benefits of federated clustering*
Federated clustering offers several benefits for M2M communication networks:

- **Enhanced security:** By decentralizing the clustering process, federated clustering reduces the risk of single points of failure and makes it more difficult for attackers to compromise the entire network.

- **Data privacy:** Since raw data is not transferred during the federated learning process, federated clustering preserves data privacy and complies with data protection regulations.

**Figure 1 Proposed architecture.**

- **Scalability:** Federated clustering is inherently scalable, as it distributes the computational load across multiple nodes and reduces the need for centralized coordination.
- **Efficiency:** By leveraging local computations and reducing the volume of data exchanged between nodes, federated clustering improves the efficiency and reduces the communication overhead of the network.

### *Implementation of federated clustering*

The implementation of federated clustering in the proposed framework involves several key components:

- **Local clustering algorithms:** The framework utilizes local clustering algorithms, such as K-means or density-based spatial clustering of applications with noise (DBSCAN), to group nodes based on their interactions and trust values.
- **Cluster head selection mechanism:** A selection mechanism is employed to identify the most suitable nodes to serve as CHs. This mechanism considers factors such as trust values, computational capabilities, and energy levels.
- **Federated learning framework:** A federated learning framework is used to aggregate and update models from CHs. This framework ensures that data privacy is preserved and that the global model is continuously improved.
- **Communication protocols:** Efficient communication protocols are implemented to facilitate the exchange of models and updates between CHs and the central aggregator. These protocols minimize communication overhead and ensure timely updates.
- **Security measures:** The framework incorporates security measures, such as encryption and authentication, to protect the integrity and confidentiality of the data exchanged during the federated learning process.

Initially, wireless M2M machines are capable of communicating with each other and continuously seeking services in a real-time environment. In this setup, the machines are clustered by adopting federated learning (FL) to improve data privacy and machine secrecy. For clustering, a local model is generated based on individual machine factors such as distance, energy, density, degree, and stability. This local model is then encoded using Lagrange encoding, and the coded data is privately shared with the edge server using theoretical information. After manipulating the algebraic structure of the coding, FL implements Lloyd's K-means algorithm on the coded data to acquire clustering results. Once the machines are clustered, the cluster head is selected.

We have intensified throughput and energy efficiency, thus minimizing overhead and premature CH death, by implementing dual CH selection. CH selection is based on multiple parameters such as distance, energy, capacity, feedback, density, packet delivery ratio (PDR), and trust. The machines in each cluster are ranked on the basis of these parameters. The machine ranked first is selected as the CH, and the machine ranked second becomes the sub-CH. As a result, intra-communication is performed by the sub-CH, where M2M communication inside the cluster is executed by the sub-CH. Similarly, inter-communication is accomplished by the CH, involving CH-to-CH communication

and CH-to-edge server communication. This approach improves the throughput and energy efficiency of M2M communication.

### Lloyd's K-means algorithm

Lloyd's K-means algorithm, a widely used local search technique for clustering, has been a staple in many ML studies due to its effectiveness and simplicity. The algorithm starts by selecting $k$ random centers from the dataset, where $k$ represents the number of clusters to be formed. These initial centers are chosen randomly from all samples, and then the samples are assigned to one of these $k$ clusters based on their proximity to these centers.

Lloyd presented a local search technique for clustering K-means. Since it is one of the most widely used algorithms, most ML studies employ it as their primary clustering method. Beginning with k randomly chosen centers, Lloyd's approach selects k samples at random from all of the samples and allocates them to k clusters or data sets at random. Relocate the centers to the centroid of the newly created clusters after allocating each sample to the closest center. Assign and recompute the center until convergence is achieved by repeating these two procedures. The Algorithm 1 has described the complete process.

---

**Algorithm 1:** Lloyd's K-means Algorithm

**Input:** data matrix $Z \in \mathbb{S}^{a \times m}$, cluster number k
**Output:** Clustered data points
```
/* Initialization                                                      */
```
1  Randomly choose k centers;
2  **while** *no convergence* **do**
```
       /* Step 1                                                       */
```
3     According to Equation 2, update the indication matrix $Y$;
4     Allocate each point to the nearest center;
```
       /* Step 2                                                       */
```
5     Assign the k centers to the new clusters' centroid;
6     Update matrix $X$ in Equation 2;

---

### K-means algorithm

Given a data matrix $Z \in S^{a \times m}$ that consists of m samples $z_1, z_2, \ldots .. z_n$. The goal of the K-means clustering problem is to find a partition $\pi = \{\pi_1, \pi_2 \ldots \ldots \pi_k\}$ such that the total square distance between each point and its nearest center is as little as possible. We can formulate the problem as:

$$X^{-} \sum_{i=1}^{k} \sum_{z_i \in \pi_i} ||z_i - x_i||_2^2 \tag{1}$$

where $x_i$ denotes the centroid of the cluster $\pi_i$. Considering the definitions of indicator matrix Y and squared Fronten is norm. Afterwards, the K-means clustering issue may be expressed as

$$\min_{Y \in Ind, X} ||z - XY^S||_Y^2 \tag{2}$$

where $X = [x_1, x_2, \ldots.x_k] \in S^{a \times m}$ is the center of the k-clusters, Y is the indicator matrix, and $Y \in S^{m \times k}$.

## Unassailable channel selection & channel division

After successful clustering, we have selected the secure and optimal channel for performing secure communication. For that purpose, we have implemented a QDOA that designates the secure and optimal channel based on channel occupancy, signal-to-noise ratio (SNR), angle, capacity, channel state information (CSI), bit rate, and feedback. Here, to reduce latency, the CH and edge server select the secure channel collaboratively. Furthermore, the secure channels are split by executing channel division. The entire secure channels are fragmented into three divisions such as, $C_1, C_2$, and $C_3$;

- $(C_1)$: channel utilized during cluster formation stage for broadcasting.
- $(C_2)$: channel used for intra-communication.
- $(C_3)$: channel adapted for intercommunication.

By dividing the channel into multiple fragments, the overhead and latency is minimized, and the misbehavior of the machine can be identified easily, thus enhancing data privacy and security.

### *Quality diversity optimization algorithm*

Quality-diversity (QD) optimization algorithms tackle a unique set of problems in the realm of optimization. Contrary to traditional methods that aim for the optimum of a cost function, QD algorithms provide a spectrum of effective solutions differentiated by a few user-defined characteristics. These characteristics are chosen based on their relevance and significance to the user, who might be influenced by factors like design simplicity or manufacturing feasibility. Users can then select the most appealing solutions from this diverse pool, guided by their own expertise and the insights gained from the variety of solutions. This approach is especially useful in understanding the interplay between different solution attributes and their impact on performance. QD algorithms thus offer a broader perspective, enabling users to explore a wide range of effective solutions rather than focusing on a single optimal point.

$$e_\theta, a_\theta \leftarrow e(\theta). \tag{3}$$

This equation represents the core concept of QD optimization, where $e_\theta$ and $a_\theta$ signify the evaluation and attributes of the solutions based on the parameters $\theta$. The function $e(\theta)$ encapsulates the evaluation or generation of solutions, reflecting the diversity-centric approach of QD algorithms.

It is assumed that the objective function yields a behavioral descriptor (or feature vector) $a_\theta$ in addition to the fitness value $e_\theta$. While the fitness value $e_\theta$ measures the solution's effectiveness, the behavioral descriptor (BD) typically explains how the solution addresses the issue.

Looking ahead, the objective function is expected to be maximized while also maintaining generality. The aim of QD optimization is to identify the parameters $\theta$ that yield the highest

fitness value for each point *a* in the feature space *A*. This can be defined as:

$$\forall a \in A \qquad \theta^* = argmax_\theta\, e_\theta \tag{4}$$

where $a = a_\theta$. The equation represents the goal of QD optimization to find the optimal parameters $\theta$ that maximize the fitness value $e_\theta$ for each distinct behavioral descriptor *a* within the feature space *A*. This approach enables a comprehensive exploration of the solution space, taking into account both the effectiveness of solutions and the diverse ways they address the problem.

Upon initial observation, QD algorithms might appear akin to multitask optimization, as they seem to address an optimization issue for every possible combination of features. The QD problem is essentially a series of optimizations, each constrained by a specific BD. This is particularly challenging because, firstly, the feature space *A* could be continuous, leading to an infinite number of problems, and secondly, the BD is unknown prior to the application of the fitness function.

A key aspect of QD algorithms is the collective approach to these numerous optimization problems, which is often more efficient than conducting separate constrained optimizations. The sharing of information between optimizations is beneficial, especially since solutions with similar feature descriptors are likely to perform well.

The effectiveness of a QD algorithm is measured by two primary criteria:

1. The quality of the resultant solution for each type of solution reflects the degree of optimization achieved.
2. The coverage of the feature space indicates how extensively the behavior space is explored.

The outcome of QD optimization is a diverse set of solutions. This collection, also known as a "collection", "archive", or "map", expands, evolves, and improves throughout the optimization process. Each point in this collection represents a distinct "species" or "type" of solution, offering a varied perspective on potential solutions.

Traditionally, evolutionary algorithms focus predominantly on the fitness value or quality for decision making, often overlooking the global optimum and the diversity of solutions. In contrast, QD algorithms consider multiple factors to explore the behavior space (diversity) more thoroughly. The QD-optimization algorithm, encompassing *I* iterations, is detailed in Algorithm 2, showcasing its approach to balancing both quality and diversity in solutions.

## Trust evaluation & secure communication

Once a secure channel is selected and divided, trust evaluation is implemented to ensure data privacy. This process is carried out by monitoring agents (MA) present in individual edge servers, which monitor the machine and evaluate its trust. The trust of all machines is determined by quantifying four parameters, including direct trust (DT), indirect trust (IDT), recent trust (RT), and data packet delivery. Direct trust is estimated through the communication and behavior of individual machines, while indirect trust is derived from the opinions of neighboring nodes with high trust. Recent trust is computed based on DT and IDT, and the trust of data packet delivery is determined by the ratio of total data

---

**Algorithm 2:** QD-Optimization algorithm (I iterations)

1  $C \leftarrow \emptyset$;
2  **for** $iter = 1 \rightarrow I$ **do**
3      **if** $iter == 1$ **then**
4          $R_{\text{parents}} \leftarrow \text{random}()$;
5          $R_{\text{offspring}} \leftarrow \text{random}()$;
6      **else**
7          $R_{\text{parents}} \leftarrow \text{selection}(C, R_{\text{offspring}})$;
8          $R_{\text{offspring}} \leftarrow \text{variation}(R_{\text{parents}})$;
9      **foreach** $\theta \in R_{\text{offspring}}$ **do**
10         $\{e_\theta, a_\theta\} \leftarrow e(\theta)$;
11         **if** $ADD\_TO\_CONTAINER(\theta, C)$ **then**
12             UPDATE_SCORES(parent($\theta$), Reward, C);
13         **else**
14             UPDATE_SCORES(parent($\theta$), -penalty, C);
15     UPDATE_CONTAINER(C);
16 **return** $C$;

---

packets received to those transmitted by the machine, including packets forwarded and dropped.

The evaluated trust values of individual machines are stored in the blockchain to enhance security. To further improve M2M communication, multiagent reinforcement learning (MARL) is exploited, with each agent trained using a deep deterministic policy gradient (DDPG) network. Additionally, an AMARLCAT is proposed to minimize latency and nonscalable convergence in data transmission. For enhanced security in M2M communication, a homomorphic encryption mechanism is employed, encrypting the data for transmission through the secure channel.

### Reinforcement learning

Automation in machines is predominantly based on reinforcement learning (RL), as illustrated in Fig. 2. In this framework, agents operate within an environment guided by intelligent rules and a reward system. The agents engage in a balance of exploitation and exploration, learning about their environment through continuous trial and error in their actions. This process of exploration and exploitation continues until the agents become adept at performing the relevant activities, effectively adapting their behavior to maximize the cumulative reward or achieve specific objectives within the environment.

### Multi-agent reinforcement learning

To complete a task faster, MARL uses autonomous, interactive systems operating in a shared environment. As a result, agents with centralized training carry out dispersed tasks. Applications for MARL systems are found in many different fields, such as distributed control, network administration, game creation, and decision support systems.

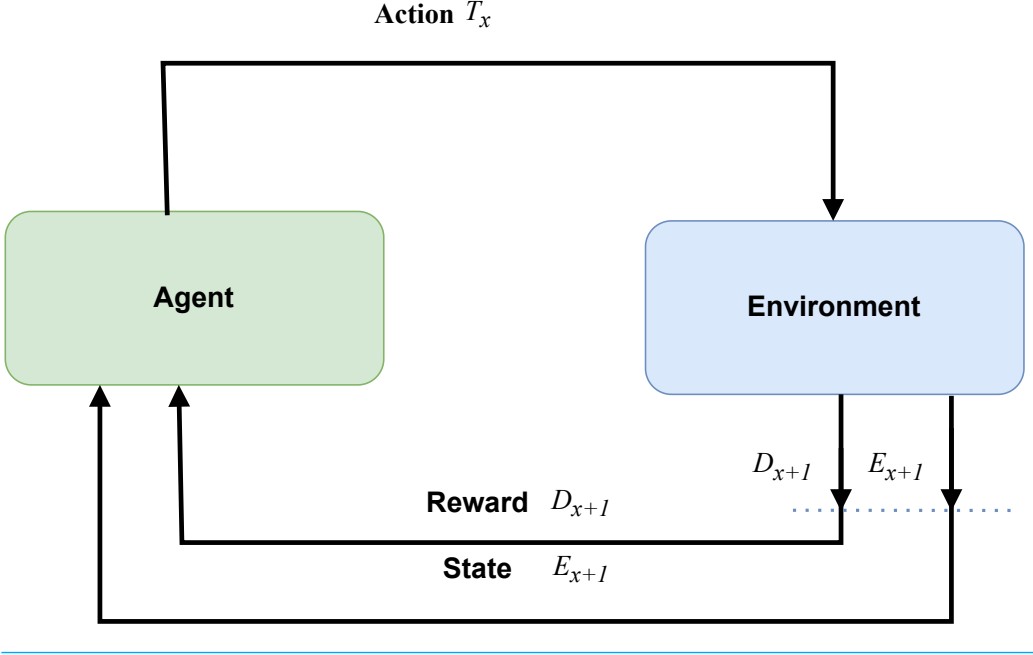

**Figure 2  Reinforcement learning (RL).**

### Deep deterministic policy gradient

The DDPG algorithm tackles the continuous action space reinforcement learning problem. Its main procedure includes: first, storing the experience data generated as the agent interacts with the environment in the memory of past experiences. Additionally, the actor-critic architecture is utilized to obtain and preserve the sampled data, leading to the determination of the optimal course of action. Figure 3 illustrates the structure of the DDPG algorithm.

In the DDPG approach, which is based on the deterministic policy gradient, the DL method is applied to a neural network that replicates the Q and policy functions. The DDPG algorithm, underpinned by the actor-critic architecture, maintains the organizational structure of the deep Q-network (DQN) algorithm. The actor and critic modules in DDPG can use the online network and target network structures, combining these with the actor-critic approach in the DQN method.

During training, the agent in state C selects Action A using the current actor-network. It then calculates the current action's $P$ value and the expected return value, $x_j = T + \gamma P'$, based on the current critic network. The actor target network selects the best action $A'$ based on past learning experiences, while the critic target network calculates the $P'$ value of the subsequent action. The online network parameters of the relevant module are frequently updated to reflect changes to the target network's parameters.

To update the target network parameters, DDPG employs a "soft" approach, meaning each change to the network parameters is made with very little magnitude, which enhances the training process's stability. The update coefficient is represented by $\tau$, and the "soft"

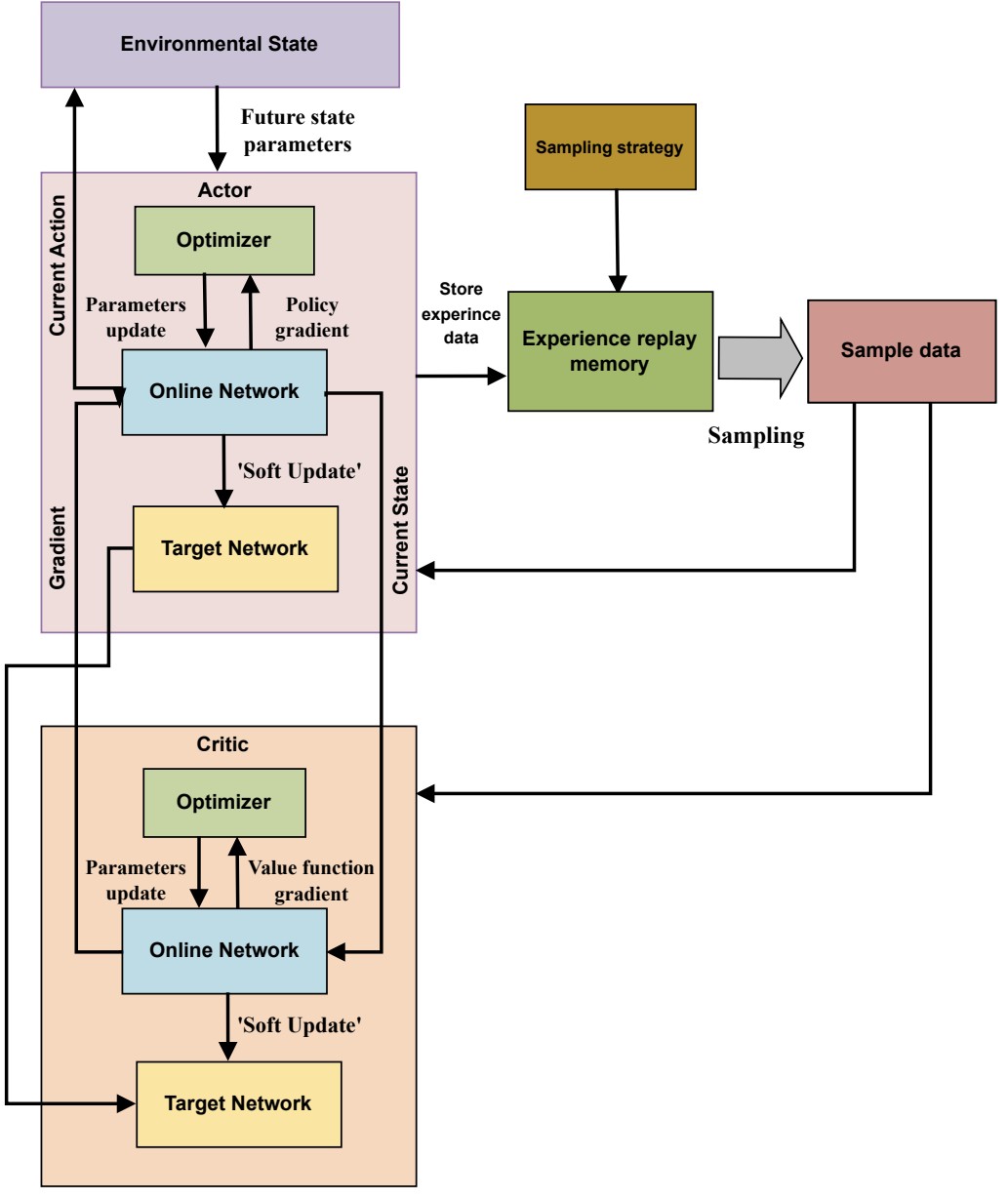

**Figure 3** The structure of DDPG algorithm.

update method can be written as Eq. (5):

$$\omega' = \tau\omega + (1-\tau)\omega' \theta' = \tau\theta + (1-\tau)\theta'. \tag{5}$$

Using the deterministic policy $\pi$, DDPG determines the action to be implemented. The objective function, which approximates the state-action function, is represented using a value network. This function is defined as the cumulative reward with a discounted factor, as shown in Eq. (6):

$$K(\theta) = B_\theta \left[ e_1 + \gamma e_2 + \gamma^2 e_3 + ... \right]. \tag{6}$$

In this equation, $K(\theta)$ represents the objective function with respect to the policy parameters $\theta$. The term $B_\theta$ reflects the baseline function or the expected value under the policy $\pi$. The elements $e_1, e_2, e_3, \ldots$ denote the sequence of rewards obtained from the environment, and $\gamma$ is the discount factor, which quantifies the importance of future rewards compared to immediate ones. This formulation allows DDPG to effectively balance immediate and long-term rewards, optimizing the policy towards more beneficial outcomes over time.

Equation (7) delineates the updating process for the critic's online network's parameters, focusing on minimizing the mean square error of the loss function. This equation is essential in the training phase, where the goal is to optimize the critic network to accurately predict the value of state-action pairs:

$$K(\omega) = \frac{1}{n} \sum_{k=1}^{n} (x_k - P(\varnothing(C_k), A_k, \omega))^2. \tag{7}$$

In this equation, $K(\omega)$ represents the loss function with respect to the critic network parameters $\omega$. The term $x_k$ is the target value for the $k$-th data sample, and $P(\varnothing(C_k), A_k, \omega)$ is the prediction of the critic network for the $k$-th state-action pair $(C_k, A_k)$. The loss function computes the average of the squared differences between the target values and the network's predictions over $n$ samples. Minimizing this loss function helps calibrate the critical network to estimate the value function associated with the policy better.

Equation (8) illustrates how the network parameters for the actor online network are adjusted based on the policy's loss gradient.

$$\nabla_k(\theta) = \frac{1}{n} \sum_{k=1}^{n} \left[ \nabla_a P(C_k, a_k, \omega) \nabla_{\theta_\pi} \right]. \tag{8}$$

### Adaptive multi-agent reinforcement learning for context-aware transmission

The AMARLCAT system, when compared to advanced DDPG-based MARL systems, demonstrates significant improvements in reducing convergence time and enhancing converged performance. Additionally, it surpasses traditional MAC methods in performance. One notable advantage of AMARLCAT is its adaptability, offering various QoS options. Extensive simulations have been conducted to evaluate different state space definitions and metrics, with AMARLCAT showcasing scalability and resilience in all tested scenarios.

The foundation of most reinforcement learning algorithms is the idea of an iterative approach. These algorithms use what are known as temporal-difference updates to spread knowledge about state-action pairings, $Q(s, b)$, or values of states, $V(s)$. The basis for these updates is the difference between the two temporally distinct estimations of a certain state or state-action value. It updates state-action values in the environment after each real transition, $(s, b) \to (s', r)$, using the following formula:

$$Q(s, b) \leftarrow Q(s, b) + \delta[r + \gamma Q(s', b') - Q(s, b)]$$

where the discount factor $\gamma$ and the rate of learning $\delta$ are given. Once an action is completed and the environment returns a reward $r$, it moves to a new state $s'$, and action $b$ is selected in state $s'$. This changes the value of acting in state $s$.

Balancing discovery and utilization is key to developing effective reinforcement learning agents. It seeks to provide an equilibrium between the exploiting of the agent's knowledge and the enriching exploration of the agent's knowledge. Commonly, an agent will usually behave greedily, but it will choose an action at random with a chance of $\epsilon$. This kind of behavior is known as $\epsilon$-greedy. Reduce $\rho$ over time to get the most out of both exploration and exploitation.

There are two main ways that reinforcement learning is used in multi-agent systems: joint action learners or multiple individual learners. Previously, single-agent reinforcement learning algorithms were used to deploy numerous agents for each task. In the latter case, an agent watches the activities of other agents and communicates its actions to the others. This approach comprises specialized algorithms designed for multi-agent scenarios, considering the interactions between multiple agents.

The environment seems to be dynamic since the likelihood of transition when taking action in a state varies over time when multiple individual learners believe any other agents to be a part of the environment. Joint action learners were created to expand their value function and take into account each state's potential combination of actions by elements to overcome the illusion of a dynamic environment. The number of values that need to be computed increases exponentially with each new addition to the system due to the joint action consideration. Therefore, our study focuses on many multiagent learners since we are interested in scalability and minimum communication latency between agents.

The AMARLCAT system comprises two primary components: setting up a database and the modification of the transmission overhead $m$. The first step involves obtaining the best transmission policy under the current transmission overhead by inputting $m$ and the state after training. This process leads to the creation of an offline database, where the transmission overhead $m$ range is determined using empirical values. Each $m$ is trained to collect and store the relative DDPG network attributes $\theta$. Constructing the database requires training $M$ events for every $n$ value, making it a time-intensive process.

However, the second stage of training offers significant time savings. Initially, a starting transmission overhead of $M_0$ is defined, and the corresponding network parameters $\theta$ are retrieved from the stage 1 database. The contention process is then simulated to determine the optimal $n$. Specifically, after completing single-agent training, all terminals use the existing model parameters for multi-agent training. During this phase, if a terminal detects an idle channel, it sends data with a probability determined by its current instantaneous DDPG output. Following each transmission period, the control unit computes the mean reward and modifies the communication cost according to success/failure signals. The transmission overhead is significantly increased following $p_{fail}$ consecutive collisions but modestly decreased after $p_{idle}$ consecutive idle intervals.

The methodology outlined in Algorithm 3 starts with $M$ terminals, where $\rho$ regulates the number of terminals per iteration, and $M_{targets}$ is the number of competing terminals.

---

**Algorithm 3:** AMARLCAT

---

```
/* Phase 1: Independent Training for Single Agents        */
/* Setup                                                   */
```

1 Agents: Initialize parameters $\Theta_n(\text{n=1},\dots,N)$ using a normal distribution;

2 Control Unit: Define $m_{\max}$ and $m_{\min}$ as upper and lower transmission fees;

3 Assign $m_{\min} = 500$, $m_{\max} = 10000$, $N_{\text{target}} = \rho N$;

4 $p_{\text{tx}} = \min\left( \sqrt{\frac{2\delta}{T_s N_{\text{target}}^2}}, \frac{1}{N_{\text{target}}} \right)$;

5 **for** $q = 1$ **to** $M$ **do**

6     **for** $n = 1$ **to** $N$ **do**

7        Perform training of DDPG respecting agent-$n$ with current $m$, updating $\Theta_n$;

8        Record the association between $m$ and $\Theta_n$ in a database;

9     Increment $m$ by $m_i$;

```
/* Phase 2: Collaborative Multi-Agent Training            */
/* Setup                                                   */
```

10 Agents: Reset parameters $\Theta_n(\text{n=1},\dots,N)$ to those from initial training with $m_0$;

11 Control Unit: Set initial transmission fee as $m_0$;

12 **for** *event* $= 1$ **to** $T$ **do**

13     **for** $n = 1$ **to** $N$ **do**

14        **if** *Agent-n detects an idle channel and DDPG suggests transmission* **then**

15           Agent-$n$ transmits with probability $p_{\text{tx}}$ in the current time slot;

16        **else**

17           Agent-$n$ remains silent in the current time slot;

18        $R_k =$ Compute the average utility for all agents;

19        **if** *Sequential transmission failures reach $q_{fail}$* **then**

20           Increase $m$ by $m_u$;

21        **else**

22           **if** *Channel sensed idle $q_{idle}$ times consecutively* **then**

23              Decrease $m$ by $m_d$;

24           **else**

25              Keep $m$ constant and proceed to the next event;

---

## Surveillance & maintenance

Eventually, communication surveillance and maintenance are implemented to amplify security, thereby minimizing the packet loss rate. For that purpose, at first, the MA will continuously monitor the network based on three bases: initiating communication threat, vulnerability analysis, and link state prediction. The initializations of communication threats indicate the direct transmission of data from one cluster member (CM) to another CH or CM. Then, the vulnerability analysis is established by analyzing the traffic features of data transmission to identify anomalies. Following that, the link state prediction is

accomplished to minimize the packet loss. Here, the link failure is predicted in the next hop by the received signal strength, energy, response time, and quality of each machine, which aids in accurate link breakage identification. If the MA identifies any one of these threats or failures in any of these three bases, then it performs three operations. Communication from certain machines that initiate threats or vulnerabilities will be blocked if the identified base is a communication thread or anomaly.

Communication from certain machines that initiate threats or vulnerabilities will be blocked if the identified base is a communication thread or anomaly. Furthermore, if the MA identifies link failure, then it executes two operations as re-direct or new secure channel selection.

Once link breakage is identified, the MA checks whether there is any free secure channel that already completed its data transmission and re-directs the data to that secure channel. Otherwise, if the communication in the entire secure is occurring, then the new secure channel is identified. By performing these processes, the security of data transmission is amplified, thereby minimizing packet loss.

### *Homomorphic encryption*
Homomorphic encryption (HE) is an encryption method enabling certain computations on encrypted data without decryption for specific mathematical operations to be performed on ciphertexts. This process results in the generation of another ciphertext. Importantly, the outcome of these operations on the encrypted text mirrors that of operations performed directly on the plaintext. This gives the effect of having conducted the operations on the plaintext itself without any modification or distortion. The use of this encryption method enables users to interact with encrypted data without needing access to the actual content from the sender or the public key for decryption. HE is particularly valuable in preserving privacy across various applications, including cloud data storage, and in enhancing the security and transparency of elections. Furthermore, it addresses challenges in maintaining the privacy of processes and stored data in databases.

## Blockchain implementation for trust management
Our blockchain implementation for trust management in M2M communication is based on a Merkle tree structure, which provides efficient and secure verification of trust data.

### *Merkle tree-based blockchain structure*
The blockchain consists of blocks, each containing:

- Block header:
  - Previous block hash
  - Merkle root
  - Timestamp
  - Nonce

- Transactions: Trust evaluation events

  Each transaction in a block represents a trust evaluation and includes:

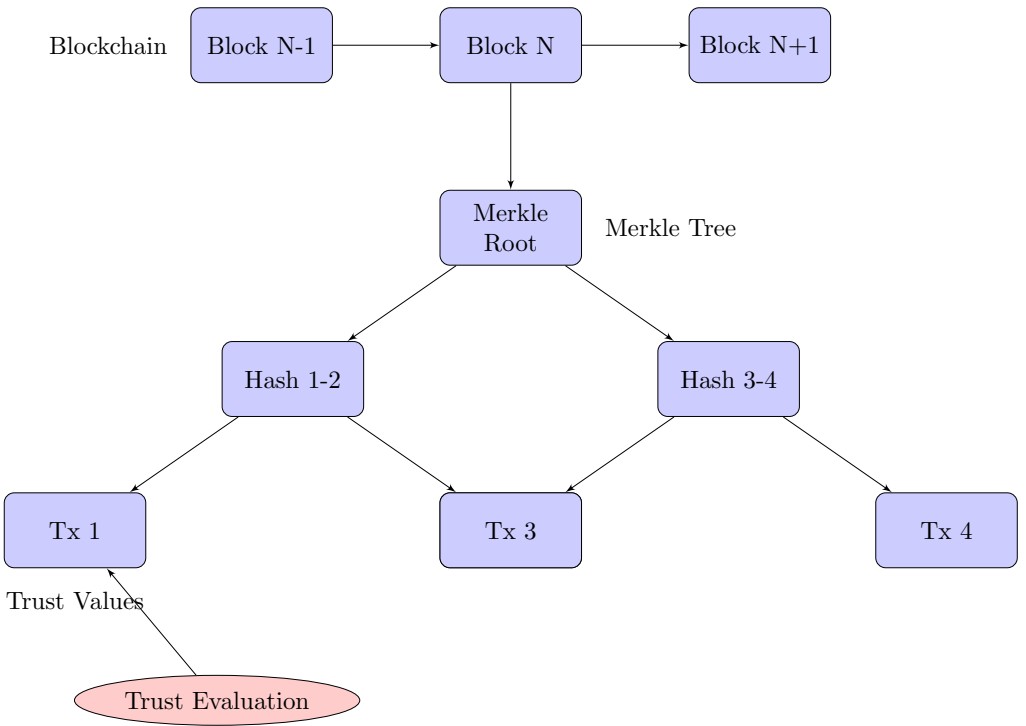

**Figure 4** **Merkle tree-based blockchain for trust management in M2M communication.**

- Evaluator ID
- Evaluated ID
- Trust value
- Timestamp
- Evaluation metrics

### *Merkle tree construction*

Figure 4 exhibits the blockchain structure for trust management:

1. Each transaction is hashed using SHA-256. 2. These transaction hashes form the leaves of the Merkle tree. 3. Pairs of leaf nodes are then hashed together to form parent nodes. 4. This process continues until a single hash (the Merkle root) is obtained.

The Merkle root, which is part of the block header, offers a concise summary of all the transactions contained within the block.

### *Trust value storage and verification*

When a machine calculates a new trust value:

1. It creates a transaction and broadcasts it to the network. 2. Nodes validate the transaction and add it to the current block. 3. Once enough transactions are collected or a time limit is reached, a new block is created. 4. The Merkle tree for the block is constructed, and the Merkle root is calculated. 5. The block is then added to the blockchain through the consensus mechanism.

### Efficient trust value retrieval

The Merkle tree structure allows for efficient verification of trust values:

1. A machine can request a specific trust value along with its Merkle proof. 2. The Merkle proof consists of the transaction and the minimum set of hashes needed to reconstruct the path to the Merkle root. 3. The requesting machine can verify the transaction's inclusion in the block by recalculating the Merkle root using the provided proof.

This process allows for trustless verification of individual transactions without needing to download the entire blockchain.

### Consensus mechanism

We employ a practical Byzantine fault tolerance (PBFT) consensus mechanism, adapted for our Merkle tree-based structure:

1. A leader node proposes a new block, including the Merkle root in the block header. 2. Validator nodes verify the Merkle root and the block's validity. 3. If a supermajority of validators approves, the block is added to the chain.

### Trust propagation and utilization

Trust propagation leverages the Merkle tree structure:

1. Machines can efficiently query and verify trust transactions for any node in the network. 2. Indirect trust is computed by traversing trust relationships, using Merkle proofs to verify each step in the trust chain.

Before initiating communication, a machine can quickly verify the trustworthiness of potential partners by requesting relevant trust transactions and their Merkle proofs.

### Security measures

The Merkle tree structure enhances security:

- Tamper evidence: Any change in a transaction would alter the Merkle root, making tampering immediately detectable.
- Efficient auditing: The Merkle structure allows for quick verification of any transaction's integrity.
- Reduced storage: Light nodes can participate in the network by storing only block headers (including Merkle roots) instead of full blocks.

### Scalability considerations

The Merkle tree structure improves scalability:

- Sharding: Each shard maintains its own Merkle tree-based blockchain.
- Cross-shard verification: Merkle proofs allow for efficient verification of trust data across shards.
- Pruning: Old transactions can be pruned while maintaining the Merkle root for historical verification.

This Merkle tree-based blockchain implementation ensures efficient, secure, and scalable storage and sharing of trust values in the M2M network. It provides a robust foundation

for decentralized trust management, capable of handling the dynamic nature of M2M communications while maintaining high levels of security and efficiency.

## Interaction of the components in SAFE-CAST

As depicted in Fig. 5, SAFE-CAST components interacts with each other as follows: M2M devices send data to the federated clustering module, as the initial input of raw data from the IoT devices into the SAFE-CAST system. The federated clustering module sends "Cluster Info" to the QDOA channel selection components with details about the formed clusters, which helps in optimizing channel selection. The federated clustering module sends meta-data to the blockchain trust management component containing information about clustering results and node information which is recorded in the blockchain for trust evaluation. The QDOA channel selection system sends "Channel Assignments" to AMARLCAT. This means that the optimized results from this system are utilized by AMARLCAT for adaptive transmission. QDOA channel selection provides "Channel Quality" information to the blockchain trust management, indicating that channel quality metrics are integral to the trust evaluation process. The blockchain trust management component sends "Trust Scores" to AMARLCAT, which likely influence the decision-making process in AMARLCAT for secure and efficient transmission. AMARLCAT sends "Transmission Data" back to the blockchain trust management, creating a feedback loop that enables the trust management system to update based on actual transmission performance.

The surveillance & maintenance module, sends "Performance Metrics" to the blockchain trust management, ensuring continuous monitoring and updating of trust scores based on observed system performance, and provides "Network State" information to AMARLCAT, enabling it to adapt its strategies based on the current state of the network.

These interactions create a comprehensive, self-improving system where the federated clustering module organizes the network efficiently, and QDOA optimizes channel selection based on the network structure. The blockchain trust management component maintains a secure, decentralized record of trust, while AMARLCAT utilizes trust scores, channel assignments, and network state to optimize transmissions. Simultaneously, the surveillance & maintenance module continuously monitors the system, providing feedback that enhances both performance and security.

## Theoretical analysis of efficiency and security

To complement our experimental demonstrations, we present a theoretical analysis of the efficiency and security aspects of our proposed system. This analysis provides insights into the expected performance and security enhancements of our approach.

### Efficiency analysis

The efficiency of our system can be theoretically analyzed in terms of power consumption and computational complexity.

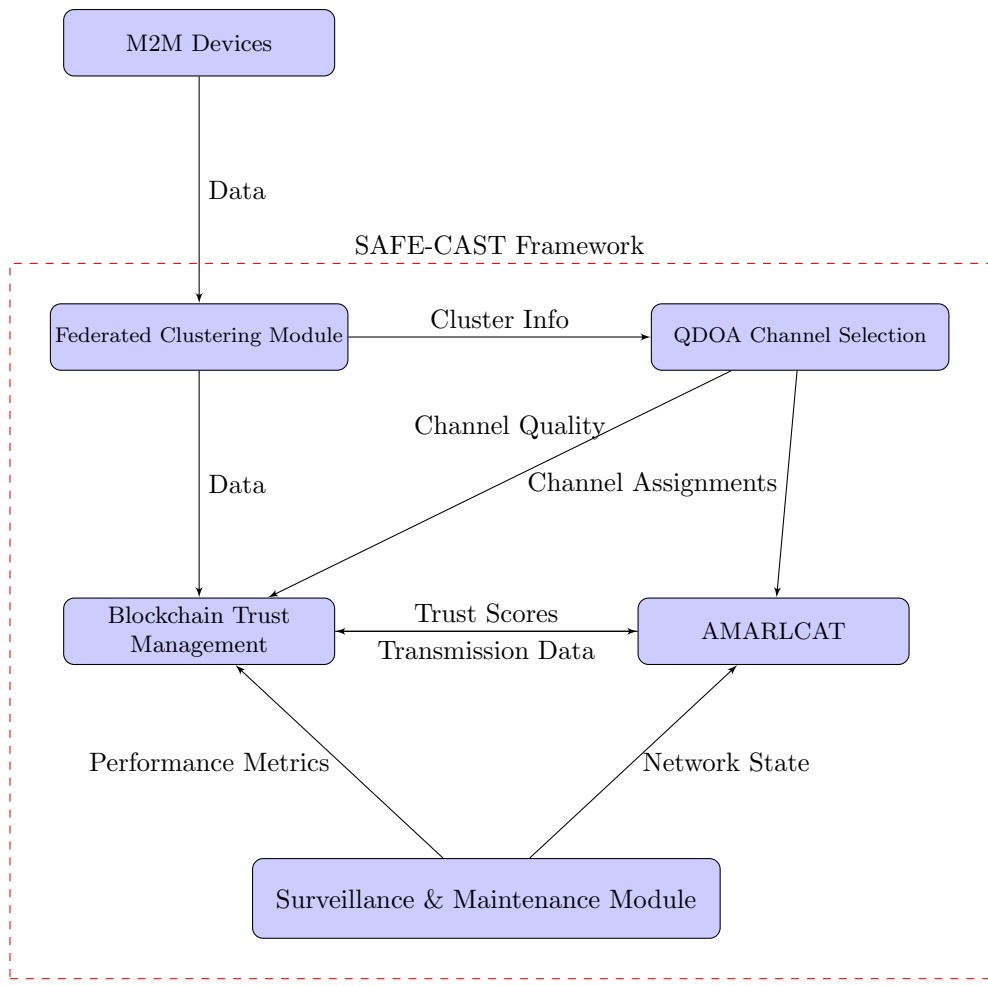

**Figure 5** **Interaction of the components in SAFE-CAST.**

*Power consumption.* Let $P_{total}$ be the total power consumption of the system, which can be expressed as:

$$P_{total} = P_{clustering} + P_{channel} + P_{communication} + P_{surveillance} \qquad (9)$$

where:

- $P_{clustering}$ is the power consumed during federated clustering
- $P_{channel}$ is the power consumed for secure channel selection
- $P_{communication}$ is the power consumed during data transmission
- $P_{surveillance}$ is the power consumed for surveillance and maintenance

Our federated clustering approach reduces $P_{clustering}$ by distributing the computational load across multiple nodes. The dual CH selection minimizes $P_{communication}$ by optimizing intra-cluster and inter-cluster communications. The QDOA for channel selection reduces $P_{channel}$ by efficiently identifying optimal channels.

*Computational complexity.* The overall computational complexity of our system can be expressed as:

$$O(total) = O(clustering) + O(channel) + O(trust) + O(surveillance) \tag{10}$$

where each term represents the complexity of the respective component, the use of Lloyd's K-means algorithm in federated clustering results in $O(clustering) = O(nkdi)$, where $n$ is the number of data points, $k$ is the number of clusters, $d$ is the number of dimensions, and $i$ is the number of iterations.

### Security analysis

The security of our system can be theoretically analyzed in terms of attack resistance and data privacy.

*Attack resistance.* Let $P(attack)$ be the probability of a successful attack. Our multi-layered security approach aims to minimize $P(attack)$ through:

$$P(attack) = P(bypass_{clustering}) \cdot P(bypass_{channel}) \cdot P(bypass_{trust}) \cdot P(bypass_{surveillance}). \tag{11}$$

Each term represents the probability of bypassing the respective security layer. By implementing secure federated clustering, channel selection, trust evaluation, and surveillance, we significantly reduce each of these probabilities, thereby minimizing the overall $P(attack)$.

*Data privacy.* The level of data privacy $DP$ can be quantified as:

$$DP = 1 - P(data\_leak) \tag{12}$$

where $P(data\_leak)$ is the probability of a data leak occurring, our use of homomorphic encryption and blockchain technology for trust storage significantly reduces $P(data\_leak)$, thus enhancing overall data privacy.

*Trust evaluation.* The accuracy of our trust evaluation mechanism can be theoretically expressed as:

$$T_{accuracy} = w_1 \cdot DT + w_2 \cdot IDT + w_3 \cdot RT + w_4 \cdot PDR \tag{13}$$

where $DT$, $IDT$, $RT$, and $PDR$ represent direct trust, indirect trust, recent trust, and packet delivery rate, respectively, and $w_1$, $w_2$, $w_3$, and $w_4$ are weighing factors.

This theoretical analysis demonstrates that our proposed system is expected to significantly reduce power consumption, minimize data leaks, and enhance overall security in edge-assisted M2M communication. The multi-layered approach to security, combined with efficient clustering and channel selection mechanisms, provides a robust framework for secure and efficient M2M communication.

## Implementation considerations

The implementation of SAFE-CAST in real-world M2M networks requires careful consideration of several factors:

1. Computational resources: While SAFE-CAST offers significant improvements, its components, such as federated learning and blockchain-based trust management, require substantial computational resources. Implementation should consider the capabilities of the target devices and potentially offload heavy computations to edge servers where possible.

2. Scalability: Our simulations demonstrate SAFE-CAST's performance with up to 100 devices. For larger networks, implementation may require hierarchical structuring or sharding techniques to maintain efficiency.

3. Integration with existing systems: SAFE-CAST is designed to be modular, allowing for phased implementation in existing M2M networks. Initial deployment could focus on the federated clustering and secure channel selection components, with trust management and advanced surveillance added in subsequent phases.

4. Regulatory compliance: Implementation must consider data protection regulations such as GDPR, particularly in the context of trust evaluation and data sharing between devices, in order to protect personally identifiable information (PII).

5. Maintenance and updates: The dynamic nature of M2M networks and evolving security threats necessitate a system for regular updates to the SAFE-CAST components, particularly the machine learning models used in threat detection and link state prediction.

These considerations ensure that SAFE-CAST can be effectively implemented and maintained in diverse M2M environments, maximizing its security and efficiency benefits.

# ANALYSIS OF KEY ALGORITHMS IN SAFE-CAST

The careful choice and integration of several critical algorithms underlie the effectiveness of SAFE-CAST. Each of the algorithms was selected for its particular strengths and appropriateness for solving the problems of M2M communication. However, it's also important to realize that each of these algorithms has its weaknesses. In this section, we will analyze in detail the reasoning behind the selection of each algorithm and its strengths and weaknesses.

## Lloyd's K-means algorithm for federated clustering

For its ease of use and effectiveness in performing unsupervised learning tasks, particularly in an M2M network context, Lloyd's K-means algorithm was selected. This algorithm is particularly computationally efficient, which is an important factor in M2M environments with many devices and large amounts of data. It also works well in a federated learning context, which is advantageous because privacy is a major concern in distributed M2M networks. However, the algorithm is sensitive to the placement of the initial centroids and may converge to suboptimal solutions. Thus, we need to pay attention to the implementation details when using this algorithm.

## Quality diversity optimization algorithm for channel selection

The channel selection problem in M2M communications needs to be solved properly. It is a task that requires an algorithm to adopt a diverse set of high-quality solutions,

maintaining the multichannel diversity vital to M2M communication in a dynamic and progressive selection of channels. One algorithm we have selected for our study is QDOA. This algorithm offers us several advantages. First, QDOA affords a set of solutions with a good degree of quality. Next, it exploits a balance between efficiently searching solution space and maintaining solution diversity. Finally, it is a method that can accommodate our problem semantically and pragmatically within the algorithmic design of QDOA.

### Blockchain for trust management

Blockchain was selected for its decentralized, tamper-proof nature—properties that are simply vital to maintaining trust in a distributed M2M network. It provides a nicely transparent and immutable record of trust evaluations, and it could enhance the reliability of trust data in an M2M environment. Moreover, it is a secure way of achieving a distributed consensus, which is another way of ''not having a central point of failure''. All this operates under the framework that M2M trust evaluations are not centralized, trust values are privacy-preserved by homomorphic encryption.

### Adaptive multi-agent reinforcement learning for context-aware transmission

AMARLCAT was developed to refine how we communicate in the rapidly changing environment of dynamic M2M (machine-to-machine) communications. It enables the adaptive and effective assembly of a dynamically transitioning network of machines—namely, the IoT devices, which acts as agents. After it learns from the agents, it becomes more efficient in the long run. We called in M2M communications suboptimal during the initial learning phase of the AMARLCAT algorithm. To make AMARLCAT problem-solving more effective over the long run requires tuning some hyperparameters during its initial setup. Hyperparameter tuning should be done according the properties of the network, *i.e.,* 5G or 6G. In our experimente we've used 6G hyperparameter tunings.

### Conclusion of the section

SAFE-CAST achieves security, efficiency, and adaptability in M2M communication by selecting and integrating algorithms that play to their individual strengths and to the preferred overall algorithmic structure. The use of federated learning with Lloyd's K-means ensures privacy-preserving clustering. A novel and enhanced QDOA provides secure, robust, and diverse channel selection. Blockchain supplies a secure, transparent trust management layer that works with our M2M communication system. AMARLCAT provides the multi-agent reinforcement learning to calculate the trust scores of the agents. The combined approach constitutes the foundation of SAFE-CAST. The algorithms—some of which are newly developed and some of which form structures not previously applied in this domain- constitute the cornerstone of our framework SAFE-CAST.

## EXPERIMENTAL RESULTS

The experimental analysis of the suggested work in enhancing privacy and communication surveillance in M2M is shown in this section. The results demonstrate the suggested

**Table 3  System specification.**

| | | |
|---|---|---|
| Hardware specification | Hard Disk | 500 GB |
| | RAM | Minimum 2 GB |
| Software specification | Simulation Tools | NS-3.26 |
| | Processor | 2.5 GHz and above |
| | OS | ubuntu-16.04 LTS (64-bit) |

**Table 4  Simulation parameters.**

| Parameters | Values |
|---|---|
| Devices/machines | 100 |
| Edge server | 2 |
| Cloud server | 1 |
| Monitor agent | 1 |
| 6 G-based base station | 1 |
| Packet size | 1,024 |
| Number of packets | 100 |
| Interval | 1.0 |
| Simulation time | 300 |
| Beam width | 6 |

strategy's outstanding effectiveness. This sub-section includes the simulation setup, comparison analysis, and research summary.

## Simulation setup

This sub-section explains the simulation setup and environment surveillance and maintenance in M2M communication. Table 3 represents the system configuration.

The proposed work is simulated by NS-3.26, Network Simulator, which is a widely used open-source discrete-event network modeling tool. The purpose of NS-3.26 is to simulate and research multiple aspects of computer networks, including Internet protocols, communication technologies, and wired and wireless networks. It is an important tool for network creation, research, and instruction. Table 4 depicts the simulation parameters.

## Hyperparameter selection and sensitivity analysis

The performance of SAFE-CAST is influenced by several key hyperparameters. In this section, we discuss the selection process for these parameters and present a sensitivity analysis to demonstrate their impact on the experimental metrics.

### *Hyperparameter selection process*

We employed a combination of grid search and manual tuning to select the optimal hyperparameters for SAFE-CAST. The primary hyperparameters and their selected values are as follows:

- Number of clusters (k) in Lloyd's K-means algorithm: 5
- Learning rate for AMARLCAT: 0.001

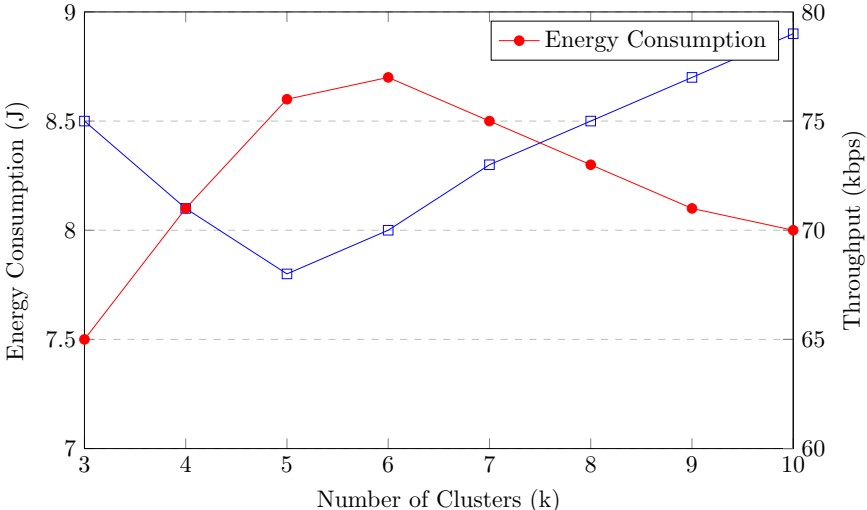

**Figure 6** **Impact of number of clusters (k) on energy consumption and throughput.**

- Discount factor ($\gamma$) for AMARLCAT: 0.99
- Number of hidden layers in DDPG networks: 3
- Neurons per hidden layer in DDPG networks: 256
- Batch size for AMARLCAT training: 64
- Trust update interval: 100 time steps

These values were chosen based on their performance across multiple evaluation metrics, including energy efficiency, throughput, security strength, latency, and packet loss rate.

### Sensitivity analysis

To understand the influence of key hyperparameters on the experimental metrics, we conducted a sensitivity analysis by varying each parameter while keeping others constant. Here, we present the results for two critical parameters: the number of clusters (k) and the learning rate for AMARLCAT.

*Impact of number of clusters (k).* We varied the number of clusters from 3 to 7 and measured its impact on energy consumption and throughput. Figure 6 shows the results of this analysis.

*Impact of number of clusters (k).* As observed in Fig. 6, increasing the number of clusters initially leads to improved energy efficiency and throughput. However, beyond $k = 5$, the benefits diminish, and energy consumption starts to increase due to the overhead of managing more clusters. This analysis justified our choice of $k = 5$ for the main experiments.

*Impact of learning rate in AMARLCAT.* We evaluated the effect of the learning rate on latency and security strength by varying it from 0.0001 to 0.01. The results are presented in Fig. 7.

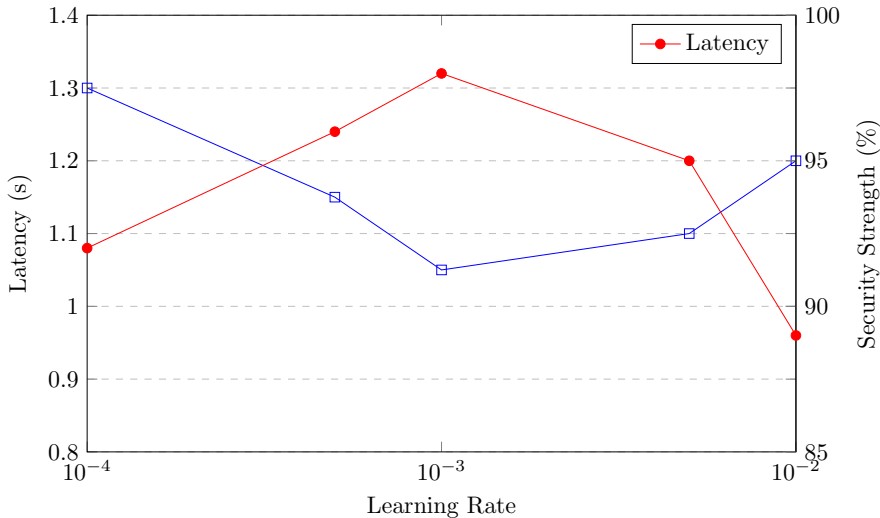

**Figure 7** **Impact of AMARLCAT learning rate on latency and security strength.**

*Impact of learning rate in AMARLCAT.* Figure 7 demonstrates that a learning rate of 0.001 provides the best balance between low latency and high security strength. Lower learning rates result in slower convergence and higher latency, while higher rates lead to instability and reduced security strength.

These sensitivity analyses provide insights into the robustness of SAFE-CAST and justify our hyperparameter choices. They also highlight the importance of careful parameter tuning in achieving optimal performance across multiple metrics.

*Impact of trust update interval.* The trust update interval determines how frequently the system updates the trust values of nodes in the network. This parameter is critical for balancing the responsiveness of the trust management system with the computational overhead. We conducted an experiment varying the trust update interval from 50 to 250 time steps and measured its impact on security strength, energy consumption, and latency.

Figure 8 illustrates the trade-offs involved in selecting the trust update interval. A shorter interval (50 time steps) results in higher security strength (99%), but also increases energy consumption (J) (8.8J) due to more frequent computations. Conversely, a longer interval (250 time steps) reduces energy consumption (7.3J), but at the cost of lower security strength (92%) and increased latency (1.4s).

*Impact of trust update interval.* Based on these results, we selected a trust update interval of 100 time steps for our main experiments, as it provides a balance between security strength (98%), energy efficiency (8.2J), and low latency (1.15s). This choice reflects our goal of maintaining high security while optimizing resource usage in M2M networks.

This experiment demonstrates the importance of carefully tuning the trust update interval to achieve the desired balance between security and performance in SAFE-CAST. It also highlights the system's flexibility in adapting to different security and efficiency requirements by adjusting this parameter.

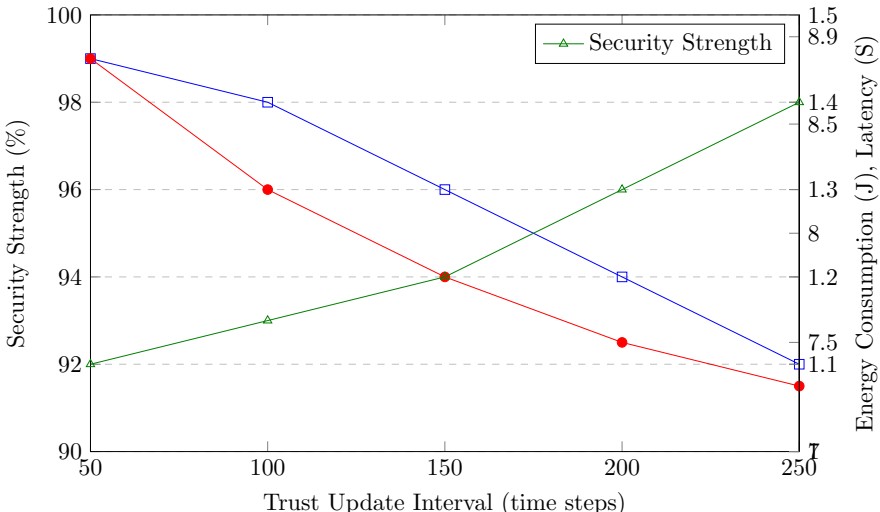

**Figure 8 Impact of trust update interval on security strength, energy consumption, and latency.**

## Comparative analysis

The proposed model is contrasted with other existing methods like event-driven duty cycling (EDDC) (*Lokhande & Patil, 2021*), long short-term memory (LSTM) (*Xu et al., 2023*), and garlic-routing-based secure data exchange framework (GRADE) (*Jadav et al., 2023*). Several metrics, including energy consumption, throughput, security strength, latency, and packet loss rate, are utilized to evaluate the suggested model graphically.

### Energy efficiency

Energy efficiency plays a pivotal role in M2M communication, significantly influencing device longevity and overall performance. To ensure prolonged operation and minimize the need for frequent battery replacements in M2M devices, the optimization of energy consumption is of paramount importance. This entails the implementation of low-power communication protocols, efficient data transmission strategies, and intelligent power management techniques.

The findings pertaining to energy consumption, as illustrated in Fig. 9 and summarized in Table 5, reveal noteworthy insights. In comparison to established methods, such as EDDC with a consumption of 10.2J, LSTM with 12.0J, and the proposed method with 8.0J, it becomes evident that the proposed approach exhibits superior energy efficiency. The advantages inherent in reduced energy consumption within the realm of M2M communication are multifaceted. It mitigates the necessity for frequent energy-related interventions, thereby curbing maintenance expenditures and extending the operational lifespan of energy-dependent devices. Moreover, diminished energy utilization contributes to a diminished environmental footprint, thereby enhancing the sustainability quotient of M2M systems.
## Energy consumption based on number of machines

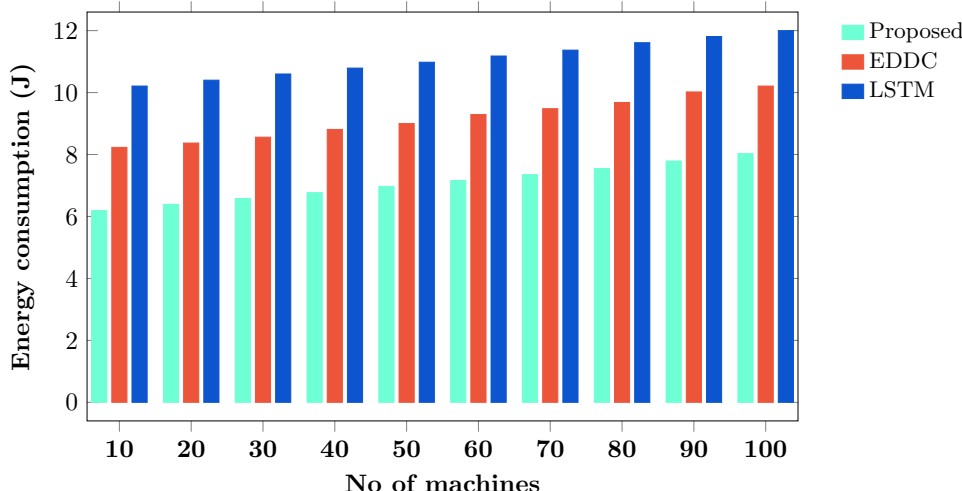

**Figure 9** Energy consumption based on the number of machines.

**Table 5 Energy consumption (J) based on number of machines.**

| No of machines | Energy consumption (J) | | |
| --- | --- | --- | --- |
| | **Proposed** | **EDDC** | **GRADE** |
| 20 | 6.4 | 8.4 | 10.4 |
| 40 | 6.8 | 8.8 | 10.8 |
| 60 | 7.2 | 9.3 | 11.2 |
| 80 | 7.6 | 9.7 | 11.6 |
| 100 | 8.0 | 10.2 | 12.0 |

**Table 6 Throughput (bits/sec) based on number of machines.**

| No of machines | Throughput (bits/sec) | | |
| --- | --- | --- | --- |
| | **Proposed** | **EDDC** | **GRADE** |
| 20 | 52,000 | 46,000 | 38,000 |
| 40 | 58,000 | 52,000 | 48,000 |
| 60 | 66,000 | 57,000 | 53,000 |
| 80 | 72,000 | 64,000 | 58,000 |
| 100 | 79,000 | 69,000 | 65,000 |

### *Throughput*

Throughput, a pivotal metric in the domain of M2M communication, quantifies the quantity of data effectively transmitted through the network within a specified time frame. It serves as a yardstick for assessing the system's capacity to handle communication requirements and gauges the efficiency of data transmission. The results pertaining to throughput are delineated in Table 6 for reference.

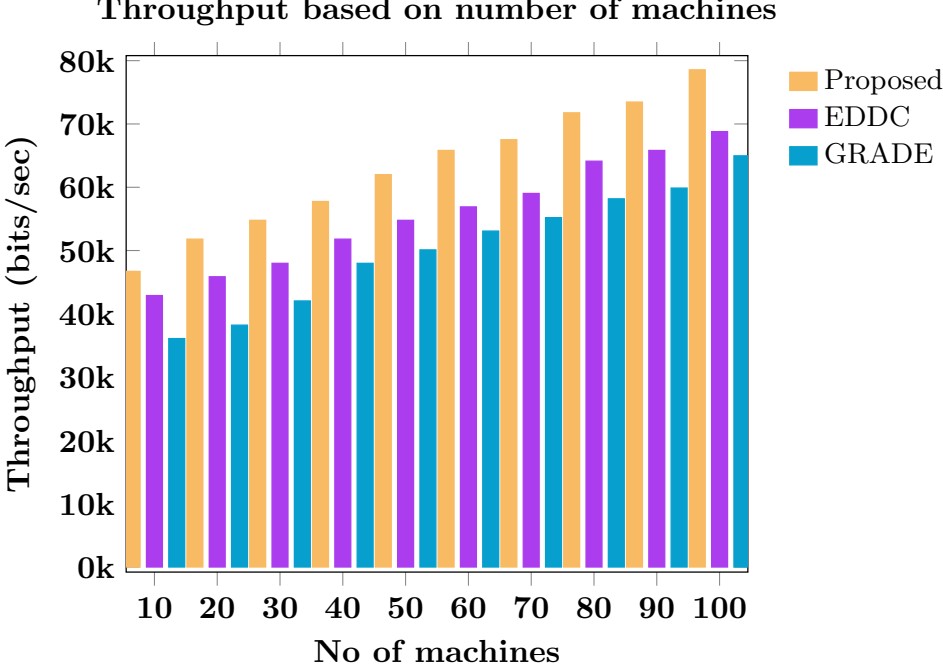

**Figure 10** **Throughput based on the number of machines.**

Figure 10 provides a graphical representation of the throughput comparison between the proposed methodology and established methods. The throughput values are as follows: 69,000 bits per second (bps) for EDDC, 65,000 bps for GRADE, and 79,000 bps for the proposed method. When juxtaposed with contemporary approaches, it becomes evident that the suggested approach achieves a superior level of throughput. This heightened throughput enables devices to engage in more rapid and responsive interactions, thereby facilitating real-time data exchange and expediting decision-making processes.

### Security strengthen

The enhancement of security in M2M communication is imperative to guarantee the confidentiality, integrity, and availability of transmitted data.

Figure 11 and Table 7 present the results of security enhancement. When compared to existing methods, the proposed approach achieves significantly higher security strength, with values of 85% for EDDC, 71% for GRADE, and an impressive 98% for the proposed method. This heightened security ensures that only authorized devices can engage in communication, mitigating the risk of unauthorized access. The implementation of robust authentication and encryption protocols guarantees the safeguarding of data during transmission, preserving both its integrity and confidentiality.

### Latency

In the context of M2M communication, latency refers to the time interval between the transmission of data from a source device to a destination device and the subsequent

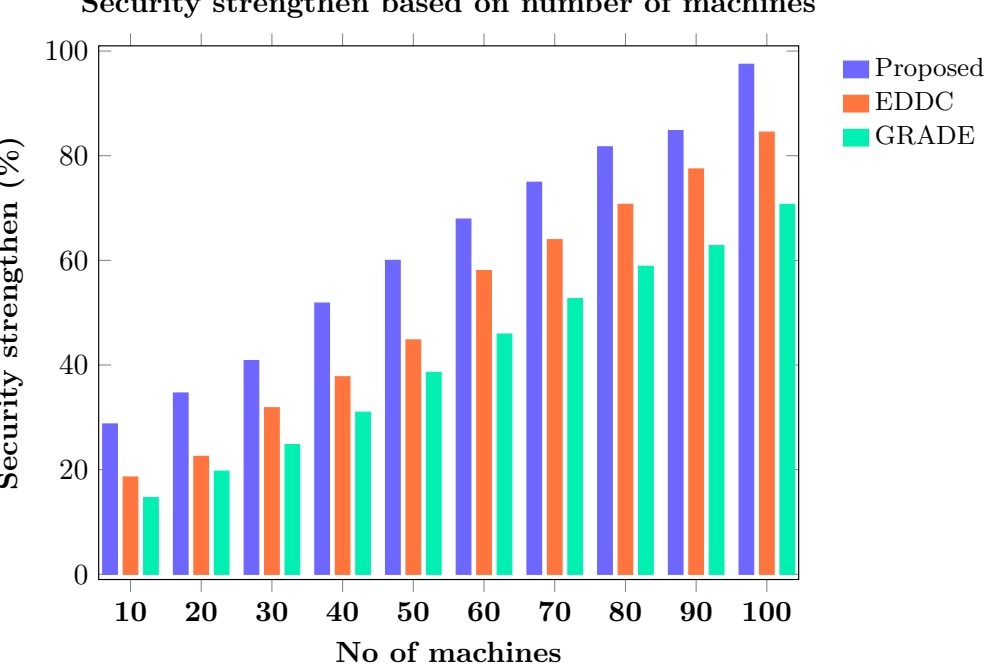

**Figure 11** Security strength based on the number of machines.

**Table 7** Security strengthens (%) based on number of machines.

| No of machines | Security strengthen | | |
| --- | --- | --- | --- |
| | **Proposed** | **EDDC** | **GRADE** |
| 20 | 35 | 23 | 20 |
| 40 | 52 | 38 | 31 |
| 60 | 68 | 58 | 46 |
| 80 | 82 | 71 | 59 |
| 100 | 98 | 85 | 71 |

reception of that data. Table 8 provides a comprehensive overview of the latency outcomes in the experimental results.

Figure 12 provides a visual representation of the latency comparison between the proposed method and existing approaches in M2M communication. The latency measurements reveal that the proposed method exhibits a lower latency of 1.09 s, as opposed to 1.65 s for EDDC and 1.70 s for LSTM. This comparison underscores the superior performance of the proposed approach in terms of reduced latency. Reduced latency has significant implications for M2M systems, as it enhances the speed and efficiency of device interactions. Lower latency facilitates faster data.

**Table 8  Latency (S) based on number of machines.**

| No of machines | Latency(S) | | |
| --- | --- | --- | --- |
| | Proposed | EDDC | LSTM |
| 20 | 0.62 | 0.7 | 0.7 |
| 40 | 0.78 | 1.0 | 1.25 |
| 60 | 0.86 | 1.2 | 1.31 |
| 80 | 0.95 | 1.42 | 1.46 |
| 100 | 1.09 | 1.65 | 1.70 |

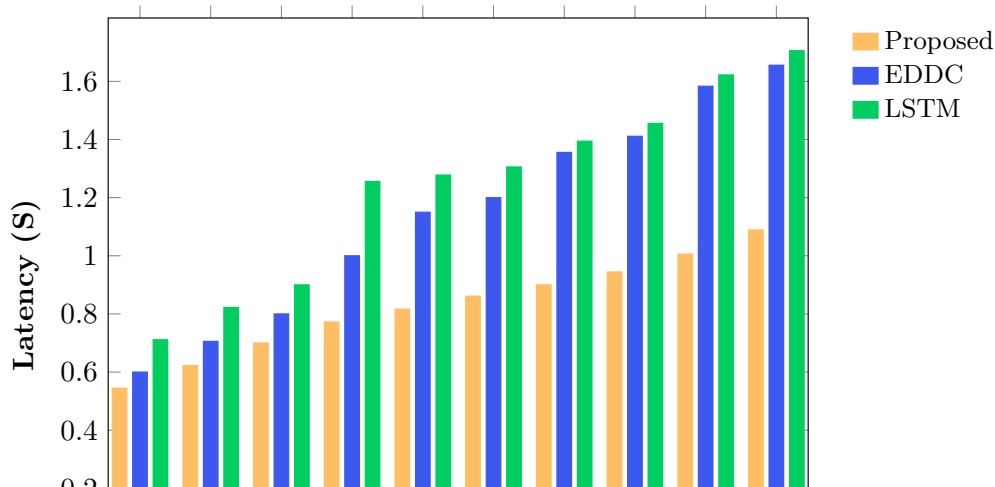

**Figure 12  Latency based on the number of machines.**

### Packet loss rate

The packet loss rate, a vital metric in M2M communication, quantifies the percentage of data packets that become lost within the network during transmission. This metric serves as a critical indicator of the reliability and quality of the communication link. Monitoring and minimizing packet loss are paramount to ensure uninterrupted data transmission and maintain the dependability of M2M systems. A low packet loss rate signifies a robust and efficient communication link, contributing to the overall effectiveness and performance of M2M networks.

Figure 13 and Table 9 present the results related to the packet loss rate. When comparing the proposed method with existing approaches, it is evident that the proposed method exhibits a considerably lower packet loss rate. Specifically, the proposed method achieves

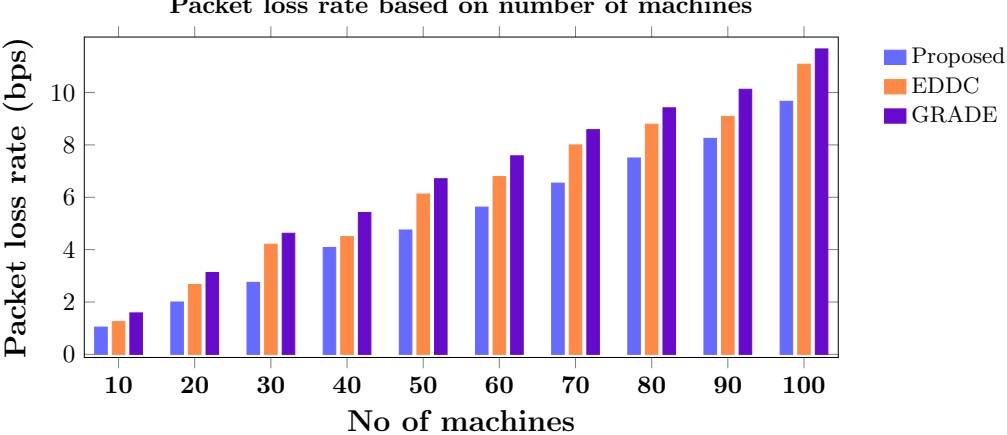

**Figure 13 Packet loss rate based on the number of machines.**

**Table 9 Packet loss rate (bps) based on number of machines.**

| No of machines | Packet loss rate (bps) | | |
|---|---|---|---|
| | **Proposed** | **EDDC** | **GRADE** |
| 20 | 1.8 | 2.4 | 2.8 |
| 40 | 3.7 | 4.1 | 4.9 |
| 60 | 5.1 | 6.2 | 6.9 |
| 80 | 6.8 | 8.0 | 8.6 |
| 100 | 8.8 | 10.1 | 10.6 |

an impressive packet loss rate of 8.8 bps, whereas EDDC records 10.1 bps, and GRADE records 10.6 bps.

A reduced packet loss rate means that a significant portion of data packets successfully reach their intended destination without experiencing loss or corruption during transit. This is of utmost importance in M2M systems, where the integrity of critical instructions and real-time data transfers must be preserved. By mitigating the likelihood of delays, inconsistent data, and potential interruptions within interconnected systems, a low packet loss rate contributes significantly to enhancing the overall performance of M2M communication.

## Framework validity

The validity of the SAFE-CAST framework is demonstrated through both theoretical analysis and extensive simulations. Our theoretical analysis, presented in 'Interaction of the components in SAFE-CAST', provides a mathematical foundation for the expected performance improvements in terms of energy efficiency, attack resistance, and data privacy. The simulation results, conducted using NS-3.26, corroborate these theoretical predictions, showing significant improvements across multiple metrics, including energy consumption (21.6% reduction), throughput (14.5% increase), security strength (15.3%

enhancement), latency (33.9% decrease), and packet loss rate (12.9% reduction) compared to state-of-the-art approaches. The consistency between theoretical predictions and simulation results, along with the comprehensive nature of our testing across various network sizes and conditions, strongly supports the validity of the SAFE-CAST framework.

### Research summary

In this study, we create a network that consists of 100 devices/machines, two edge servers, one cloud server, one MA, and one to six G-based base stations. Initially, clustering the machines by using FL based Lloyd's K-means algorithm and also selecting the two CHs. Next, select the secure and optimal channel by using the QDOA. Next, the trust evaluation is performed, and the trust is stored in the blockchain. Next, the data will be encrypted based on the homomorphic encryption mechanism, and the AMARLCAT scheme-based data transmission will be performed. Next, the channels and links are monitored. Finally, we plot the results graph for energy consumption based on number of machines, throughput based on number of machines, security strengthen based on the number of machines, latency based on the number of machines, and packet loss rate based on the number of machines. The performance of the suggested strategy is discussed in this subsection. Figures 9 to 13 shows the graphical depiction of comparative results.

## CONCLUSION AND FUTURE WORK

This study presents SAFE-CAST, a novel framework designed to enhance security and efficiency in M2M communication. Our approach integrates federated clustering, secure channel selection, dynamic trust management, and advanced surveillance techniques to address critical challenges in M2M networks. The experimental results demonstrate significant improvements in energy efficiency, throughput, security strength, latency, and packet loss rate compared to existing methods.

Although SAFE-CAST offers substantial advancements, it is important to acknowledge its limitations. The computational complexity of our approach can pose challenges for resource-constrained devices in large-scale M2M networks. Furthermore, the effectiveness of our trust management system is based on the honest participation of network nodes, which may not always be guaranteed in real-world scenarios.

SAFE-CAST can move in many promising and potentially fruitful directions to better serve the needs of its stakeholders and improve M2M network performance. The first direction SAFE-CAST can take is scalability, using nodes in very large networks. Techniques exist for using similar algorithms in hierarchical or distributed modes—which could be promising for node performance in SAFE-CAST's large- and small-scale applications. Yet, we need more research on them. Another direction is integrating emerging technologies such as 5G and 6G telecommunications networks. By using their superior bandwidth and vast node interconnectivity, future SAFE-CAST implementations could achieve better performance and a near-elimination of latency, or the problematic delay in node-to-node communication.

Ensuring strong security while maintaining privacy is still the main concern, and this work should continue to explore how to integrate post-quantum cryptography so that we

can be sure of long-term security against the sorts of new threats that are likely to arise in the next decade or so. But just as important and more immediately relevant is the question of how well SAFE-CAST will do its job in the real world, in the many maritime and smart industrial environments that are so similar to the M2M and IoT spaces we are working in and that offer so many opportunities for different sorts of efficiency gains and apparently for much better security than we have now.

Our additional experiments on the trust update interval further demonstrate the adaptability of SAFE-CAST to different network requirements. By adjusting this parameter, network administrators can fine-tune the balance between security strength, energy efficiency, and latency based on their specific needs.

Future work could explore adaptive trust update intervals that dynamically adjust based on network conditions and threat levels, further enhancing the flexibility and efficiency of SAFE-CAST in diverse M2M environments.

### AList of Abbreviations

| | |
|---|---|
| **AMARLCAT** | Adaptive Multi-Agent Reinforcement Learning for Context-Aware Transmission |
| **IoT** | Internet of Things |
| **M2M** | Machine-to-Machine |
| **QDOA** | Quality Diversity Optimization Algorithm |
| **SAFE-CAST** | Secure AI-Federated Enumeration for Clustering-based Automated Surveillance and Trust |

### Funding
This study is supported by the Turkish Scientific and Technical Council (TUBITAK) under Grant 123O739. Kasim Oztoprak is supported by the aforementioned project. The funders had no role in study design, data collection and analysis, decision to publish, or preparation of the manuscript.

### Grant Disclosures
The following grant information was disclosed by the authors:
The Turkish Scientific and Technical Council (TUBITAK): 123O739.

### Competing Interests
The authors declare there are no competing interests.

### Author Contributions
- Yusuf Kursat Tuncel conceived and designed the experiments, performed the experiments, analyzed the data, performed the computation work, prepared figures and/or tables, authored or reviewed drafts of the article, and approved the final draft.
- Kasım Öztoprak conceived and designed the experiments, analyzed the data, authored or reviewed drafts of the article, and approved the final draft.

## Data Availability

The source code is available in the Supplemental File.

## Supplemental Information

Supplemental information for this article can be found online at http://dx.doi.org/10.7717/peerj-cs.2551#supplemental-information.

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
