# Peer review of "SAFE-CAST: secure AI-federated enumeration for clustering-based automated surveillance and trust in machine-to-machine communication"

_PeerJ Computer Science, doi:10.7717/peerj-cs.2551_

## Round 0.1 · original submission · Major Revisions

I have received 3 reviews of your manuscript from scholars who are experts on the cited topic. They find the topic interesting; however, several concerns must be addressed regarding experimental setup, security analysis, and comparisons with current approaches. These issues require a major revision. Please refer to the reviewers’ comments at the end of this letter; you will see that they advise you to revise your manuscript. If you are prepared to undertake the work required, I would be pleased to reconsider my decision. Please submit a list of changes or a rebuttal against each point being raised when you submit your revised manuscript.

Thank you for considering PeerJ Computer Science for the publication of your research.

With kind regards,

Reviewer 1 ·

Basic reporting

In this work, a novel Secure AI-Federated Enumeration for Clustering-based Automated
Surveillance and Trust framework is developed to enhance M2M communication security while optimizing energy
efficiency. The organization of the paper is good, I recommend the authors to make further modification to enhance the quality of the paper

Experimental design

The experimental section must be further improved by providing figure of the simulation results.

Validity of the findings

The discussion section must be further improved by providing the merits and demerits of this approach and also provide the practical implications of this study

Additional comments

1. The beginning of the abstract does not provide enough background information to explain why such a new approach is needed and what limitations exist in existing approaches. This helps the reader to understand the motivation and importance of the research.
2.What are the main contributions of this study
3.Please clearly explain why specific algorithms were chosen, analyze the strengths and limitations of each of the algorithms.
4. In the conclusion section of the paper or in a separate future work section, clearly list several potential research directions. These directions can be based on current research shortcomings, unsolved problems, or possible extended applications.
5. All the abbreviations used mut be tabulated.
6. The motivation of this paper are not introduced in the abstract
7. Results can be further improved by adding more existing methods for comparison.
8. The manuscript should be completely proof-read

Reviewer 2 ·

Basic reporting

The basic reporting of the manuscript is clear.

Experimental design

The experiment conduction has been carried out to evaluate the metrics, namely, energy consumption, throughput, security strength, latency, and packet loss rate. Suggesting to evaluate with additional security and efficiency parameters like Network lifetime, adaptability, privacy, Resilience, Against Attacks, and so on.

Validity of the findings

It looks like some significant findings were observed. However, the findings are evaluated with only three existing models namely, EDDC, GRADE, and LSTM. Suggesting to evaluate with more existing models to justify the novelty.

Additional comments

-

Reviewer 3 ·

Basic reporting

1. The manuscript generally uses professional and clear English, but there are sections where the language could be more precise. For instance, certain technical terms are used inconsistently, which might confuse readers. I suggest reviewing the manuscript for consistency in terminology. For example, "In this paper (XXX), ...", "In this study (YYY), ...". "In this paper" should refer to your own paper.
2. The introduction provides a good overview of the challenges in M2M communication, but it could benefit from a more detailed discussion on the specific knowledge gap this study aims to address. Expanding on why existing solutions are insufficient and how this work innovatively fills that gap would strengthen the paper's impact.
3. A more detailed explanation of how each component interacts within the SAFE-CAST framework would also enhance comprehension.

Experimental design

1. The research question is well-defined, addressing significant challenges in M2M communication. However, it would be beneficial to explicitly state how this research builds on or diverges from existing methodologies. This will help to highlight the novelty and relevance of the study.
2. Some sections could benefit from additional detail, particularly in the implementation phase. For instance, more information on how the simulation environment was set up, including specific parameters and configurations, would help assess the study's replicability.

Validity of the findings

Although the manuscript claims improvements in security strength, a more detailed security analysis is warranted. This should include potential attack vectors against the proposed framework and how SAFE-CAST mitigates these threats. Additionally, a comparison with other security frameworks in similar contexts would strengthen the manuscript.

---

## Round 0.2 · Minor Revisions

All concerns raised by the reviewers have been partially addressed. However, the manuscript still needs further clarification regarding hyperparameter selection and its influence on the experimental metrics. Additional experiments are required.

Reviewer 3 ·

Basic reporting

The author addressed my concerns. Adding the existing work discussions in section 3.3 greatly improved the manuscript.

Experimental design

Could you provide more detail on how the parameters were selected? Additionally, it would be helpful to clarify how these parameters influence the experimental metrics. It may be beneficial to include further experiments to explore the impact of parameter choices more comprehensively.

Validity of the findings

The response addressed my concern.

---

## Round 0.3 · accepted · Accept

I am pleased to inform you that your work has now been accepted for publication in PeerJ Computer Science.

Please be advised that you cannot add or remove authors or references post-acceptance, regardless of the reviewers' request(s).

Thank you for submitting your work to this journal. I look forward to your continued contributions on behalf of the Editors of PeerJ Computer Science.

With kind regards,

Reviewer 3 ·

Basic reporting

Thanks for the reorganization of the manuscript. I have no concerns anymore.

Experimental design

I have no concerns about experimental design.

Validity of the findings

I have no concerns about the validity of the findings.